

# Peacock patterns and new integer invariants
# in topological string theory

**Jie Gu[1,2]⋆ and Marcos Mariño[1]†**

**1** Département de Physique Théorique et Section de Mathématiques,
Université de Genève, Genève, CH-1211 Switzerland
**2** Shing-Tung Yau Center and School of Physics, Southeast University, Nanjing 210096, China

⋆ Jie.Gu@unige.ch, † Marcos.Marino@unige.ch

## Abstract

Topological string theory near the conifold point of a Calabi–Yau threefold gives rise to factorially divergent power series which encode the all-genus enumerative information. These series lead to infinite towers of singularities in their Borel plane (also known as "peacock patterns"), and we conjecture that the corresponding Stokes constants are integer invariants of the Calabi–Yau threefold. We calculate these Stokes constants in some toric examples, confirming our conjecture and providing in some cases explicit generating functions for the new integer invariants, in the form of $q$-series. Our calculations in the toric case rely on the TS/ST correspondence, which promotes the asymptotic series near the conifold point to spectral traces of operators, and makes it easier to identify the Stokes data. The resulting mathematical structure turns out to be very similar to the one of complex Chern–Simons theory. In particular, spectral traces correspond to state integral invariants and factorize in holomorphic/anti-holomorphic blocks.

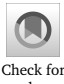

# 1   Introduction

One of the most fruitful applications of supersymmetry to geometry is the possibility to define and compute enumerative invariants based on the counting of BPS states. This idea goes back to Witten's index [1] and has led to many important developments, like the Gopakumar–Vafa invariants of Calabi–Yau threefolds [2] or the DGG index of three-dimensional manifolds [3]. More recently, through the study of their wall-crossing properties [4], it has been understood that BPS invariants can be formulated in a more analytic framework, based e.g. on the solution to a Riemann–Hilbert problem [5, 6], and closely related to WKB analysis [7]. A general mathematical construction encapsulating this idea has been put forward in [8], in which a given set of BPS invariants (or "BPS structure") is reformulated in terms of a Riemann–Hilbert problem.

In [9–12] it has been advocated that the connection between BPS invariants and analytic problems can be formulated more generally by using the theory of resurgence (see [13] for similar ideas). In this approach, the starting analytic data are asymptotic, perturbative series obtained in an appropriate quantum theory. Under some mild assumptions, the theory of resurgence associates to these series a set of complex numbers called Stokes constants. It turns out that, in some cases, the Stokes constants are integers which can be interpreted in terms of BPS counting.

Two important examples of this situation have been considered so far. The first and simpler example arises in the counting of BPS states in 4d, $\mathcal{N} = 2$ supersymmetric gauge theories. In this case, the appropriate quantum theory is obtained by putting the $\mathcal{N} = 2$ theory in the Nekrasov–Shatashvili (NS) limit [14] of the Omega-background [15]. This is equivalent to quantizing the Seiberg–Witten curve [16]. The asymptotic series obtained in this way are called quantum periods, and, as shown in [10], the corresponding Stokes constants are essentially the BPS invariants studied with the WKB method in [7].

Another example is provided by complex Chern–Simons (CS) theory on the complement of a hyperbolic knot. In this case, the asymptotic series are defined by saddle-point expansions of

the quantum invariants around classical solutions, and lead to a very rich resurgent structure. By combining different techniques, the Stokes constants associated to these series could be determined numerically in [11,12], and then conjectured analytically in closed form in many examples. Surprisingly, these Stokes constants are integers, and they turn out to be closely related to the DGG index of the three-manifold. In both types of examples ($\mathcal{N} = 2$ theories and complex CS theory), the Stokes constants are associated to singularities in the Borel plane which form infinite towers. In [11, 12] we called these arrangements, leading to infinitely many integer invariants, "peacock patterns."

In this paper we use these ideas to introduce new invariants for topological string theory on a Calabi–Yau threefold. In this case one has to address an additional difficulty which is not present in the previous examples. The natural series appearing in topological string theory are asymptotic expansions in the string coupling constant, but their coefficients are highly non-trivial functions of the moduli of the Calabi–Yau threefold. We then have to face a problem of "parametric" resurgence, similar to what happens when one studies resurgence in the large $N$ expansion of gauge theories (see e.g. [17]). In this case, the coefficients of the series in $1/N$ are non-trivial functions of the 't Hooft parameter. This observation suggests that a more manageable problem can be found if one considers, in topological string theory, the analogue of working at finite, fixed $N$ in gauge theory. From the point of view of the gauge/string duality, this means working in the small radius regime, or tensionless limit of string theory. In topological string theory, this corresponds to an expansion around the *conifold* point.

Folllowing this logic, the invariants we define are based on perturbative series in a single parameter, obtained by "quantizing" the CY moduli near the conifold point. For example, in the case of a CY with a single modulus, we obtain in this way an infinite family of perturbative series $\Phi_N(g_s)$, labeled by an integer $N = 1, 2, \cdots$. One should think about $N$ as the rank of the gauge group in a large $N$ duality (in general, each CY modulus gives an integer label). The resulting asymptotic series lead to peacock patterns, and by applying the theory of resurgence, we associate to them a (generally infinite) set of Stokes constants, which we conjecture to be integers.

Let us note that, in contrast to the usual Gopakumar–Vafa invariants, which can be obtained by rearranging the perturbative topological string expansion at large radius, the invariants we define are intrinsically non-perturbative. They encode information about the large order behavior of the perturbative series and about non-perturbative sectors which are invisible in conventional perturbation theory.

Once these invariants are defined, one should ask whether they can be effectively computed, and what is their interpretation. We show in this paper that, in the case of *toric* CY manifolds, the invariants can be calculated efficiently in many cases, and in some examples one can even obtain conjectural expressions for them in terms of $q$-series. This is done by appealing to the TS/ST correspondence, which is a conjectural equivalence between topological string theory on toric CY threefolds and spectral theory. The TS/ST correspondence was formulated in the form we need here in [18, 19], based on previous insights in [20–25]. It provides in particular a non-perturbative definition of the perturbative series $\Phi_N(g_s)$ in terms of the spectral trace $Z_N(g_s)$ of a trace class operator. We stress that our definition of the integer invariants only involves the resurgent structure of the asymptotic series and does *not* depend on the existence of a non-perturbative definition for them. However, the existence of the analytic functions $Z_N(g_s)$ makes the task of computing the invariants much easier.

In the toric case, the structure we unveil has many similarities to complex CS theory on hyperbolic knot complements. In this theory, for each complex gauge group and each hyperbolic knot one obtains a collection of perturbative series, associated to different flat complex connections in the complement of the knot. The non-perturbative object underlying these series is the state integral, or Andersen–Kashaev invariant [26–28]. The counterpart of these invariants in

the topological string are precisely the spectral traces appearing in the TS/ST correspondence. However, and in contrast to CS theory, the conventional topological string theory provides a *single* perturbative series. A resurgent analysis, like the one we will perform in this paper, unveils additional asymptotic series which are invisible in perturbation theory, and should be regarded as the "hidden" sectors of topological string theory (a similar resurgent analysis of topological string theory, albeit with very different tools, was performed in [29, 30]). The set of asymptotic series obtained in this way in the topological string should correspond to the collection of series appearing in complex CS theory.

The analogy with complex CS theory and the 3d/3d correspondence [31] is a good starting point to address the geometric and physical interpretation of the integer invariants we define. It suggests that there should be some sort of field theory dual to topological string theory, in which the integer invariants we define have a direct enumerative meaning. Another route to interpret our invariants is the rich structure of BPS/DT invariants of the underlying CY threefold. The results of [8] and the work on exponential networks in [32–36] suggests that one could use resurgent tools to compute these invariants, and we believe that the invariants we define here are simpler versions of the full BPS/DT invariants of the CY. These analogies are however not fully precise, and more work is needed to obtain a direct BPS interpretation of our invariants.

The paper is organized as follows. In section 2 we define the new integer invariants as Stokes constants of appropriate series arising in topological string theory, and we provide the necessary background from the theory of resurgence. In section 3, after a short summary of the TS/ST correspondence of [18, 19], we perform detailed analysis and computations in two well-known examples of toric CY threefolds: local $\mathbb{P}^2$ and local $\mathbb{F}_0$. In section 4 we conclude and list various problems opened by this investigation. There are three Appendices. The first one summarizes properties of Faddeev's quantum dilogarithm and of $q$-series which we use repeatedly in the paper. The second one explains how to obtain efficiently one of the relevant perturbative series in local $\mathbb{P}^2$. Finally, the third Appendix summarizes the Hunter–Guerrieri algorithm, which makes it possible to extract trans-series from the large order behavior of a Gevrey-1 series in the oscillatory case, where usual Richardson extrapolation cannot be applied.

## 2 Defining the new invariants

### 2.1 Perturbative series in topological string theory

In this section, $X$ will be a CY threefold, compact or non-compact. We will denote by $\boldsymbol{T} = (T_1, \cdots, T_r)$ its complexified Kähler moduli. In the case of toric CY threefolds, Kähler moduli can be of two types: "true" moduli, denoted by $\boldsymbol{t}$, or mass parameters $\boldsymbol{m} = (m_1, \cdots, m_s)$ (see e.g. [37]). In topological string theory, the basic objects are the genus $g$ free energies $F_g(\boldsymbol{T})$, which can be regarded as generating functions of Gromov–Witten invariants at genus $g$:

$$F_g(\boldsymbol{T}) = \sum_{\boldsymbol{d}} N_{g,\boldsymbol{d}}\, \mathrm{e}^{-\boldsymbol{d}\cdot\boldsymbol{T}}. \tag{2.1}$$

In this equation, $\boldsymbol{d} = (d_1, \cdots, d_r)$ are non-negative integers representing a two-homology class (or vector of degrees), and $N_{g,\boldsymbol{d}}$ is the Gromov–Witten invariant.

From the point of view of the global picture of the moduli space provided by mirror symmetry, the series (2.1) is an expansion near the large radius point. The moduli space is covered by overlapping open patches, each of which is parametrised by a different set of so-called flat coordinates, related by electromagnetic duality transformations in $\mathrm{Sp}(2r, \mathbb{Z})$, or changes of

frames. In each frame we have different genus $g$ free energies, related among themselves by formal Fourier transforms [38]. In addition, there are preferred choices of frame associated to particular points in moduli space. We will be interested in particular in the free energies associated to a special point in the conifold locus of moduli space. When there is a single Kähler parameter, the conifold locus is a point. In the multi-parameter, toric case, it has been noted in [19] that there exists a particular point in the conifold locus where its connected components cross transversally. We will call this point the *maximal conifold point*, as in [19, 39][1]. There is a special choice of frame and flat coordinates $\lambda = (\lambda_1, \cdots, \lambda_r)$ such that the maximal conifold point is located at

$$\lambda = 0. \tag{2.2}$$

The free energies in this frame will be denoted by $\mathcal{F}_g(\lambda)$, and we will call them *conifold free energies*. These free energies are given by the sum of a singular part $\mathcal{F}_g^s(\lambda)$ with a universal structure [40], and a regular part $\mathcal{F}_g^r(\lambda)$:

$$\mathcal{F}_g(\lambda) = \mathcal{F}_g^s(\lambda) + \mathcal{F}_g^r(\lambda). \tag{2.3}$$

The singular part has a pole of order $2g - 2$ at the maximal conifold point, for $g \geq 2$, and logarithmic singularities for $g = 0, 1$. More precisely, it has the structure

$$\mathcal{F}_g^s(\lambda) = \frac{B_{2g}}{2g(2g-2)} \sum_{i=1}^r \lambda_i^{2-2g}, \qquad g \geq 2. \tag{2.4}$$

The regular part $\mathcal{F}_g^r(\lambda)$ is of the form

$$\mathcal{F}_g^r(\lambda) = \sum_{n_i \geq 0} c_{g;n_1,\cdots,n_r} \lambda_1^{n_1} \cdots \lambda_r^{n_r}, \tag{2.5}$$

and it is an analytic function at the maximal conifold point. The coefficients $c_{g;n_1,\cdots,n_r}$ (except for a finite number) belong to an algebraic number field which depends on the geometry of the CY threefold. For example, in the case of local $\mathbb{P}^2$ (the canonical bundle of $\mathbb{P}^2$), which has a single modulus, the number field is $\mathbb{Q}[\sqrt{-3}]$, and we have

$$\mathcal{F}_0^r(\lambda) = -\frac{3V}{4\pi^2}\lambda - \frac{\pi^2}{9\sqrt{3}}\lambda^3 + \frac{\pi^4}{486}\lambda^4 + \frac{56\pi^6}{10935\sqrt{3}}\lambda^5 - \frac{1058\pi^8}{492075}\lambda^6 + \mathcal{O}(\lambda^7), \tag{2.6}$$

where

$$V = 2\,\mathrm{Im}\left(\mathrm{Li}_2\left(\mathrm{e}^{\frac{\pi i}{3}}\right)\right). \tag{2.7}$$

The expansion of the free energy around the conifold point is probably one of the less explored aspects of topological string theory. The large radius expansion (2.1) is well understood and it can be reformulated in terms of BPS invariants, as first noted in [2]. The expansions around orbifold points also have an algebro-geometric interpretation in terms of orbifold Gromov–Witten invariants (see e.g. [38]), but as far as we know, there is no direct geometric interpretation of the coefficients $c_{n_1,\cdots,n_r}$ appearing in (2.5). The conifold expansion should however play an important rôle in view of large $N$ dualities between string theory and gauge theory. In these dualities, string moduli are regarded as 't Hooft parameters in an underlying gauge theory, and perturbative gauge theory corresponds to the small radius, or tensionless limit of the string. In the case of the topological string, large $N$ dualities and the TS/ST correspondence suggest that the small radius regime corresponds to the conifold

---

[1]In the compact case there seem to exist analogues of the maximal conifold point in various examples, but points where components cross transversally might not be unique.

locus of moduli space. In addition, we should regard the conifold flat coordinates as 't Hooft parameters,[2]

$$\lambda_i = -N_i g_s, \tag{2.8}$$

where $N_i$ are non-negative integers and $g_s$ is the string coupling constant. The perturbative gauge theory regime corresponds to $N_i$ fixed and $g_s \to 0$, which means that we should consider the conifold expansion around $\boldsymbol{\lambda} = 0$. If we consider the total free energy, summed over all genera,

$$\mathcal{F}(\boldsymbol{\lambda}, g_s) = \sum_{g \geq 0} \mathcal{F}_g(\boldsymbol{\lambda})(-g_s)^{2g-2}, \tag{2.9}$$

we conclude that the analogue of a perturbative gauge theory expansion is obtained by setting (2.8) in (2.4) and (2.5). This produces a series in $g_s$ for each choice of the integers $\boldsymbol{N} = (N_1, \cdots, N_d)$:

$$\mathcal{F}_{\boldsymbol{N}}(g_s) = \sum_{g \geq 0} \mathcal{F}_g(-\boldsymbol{N}g_s)(-g_s)^{2g-2}. \tag{2.10}$$

For our purposes, the singular part of the conifold expansion is not relevant, and we will often consider the partition function, or exponential of the total free energies. This defines a family of formal power series in the string coupling constant

$$\Phi_{\boldsymbol{N}}(g_s) = \exp\left(\sum_{g \geq 0} \mathcal{F}_g^{\mathrm{r}}(-\boldsymbol{N}g_s)(-g_s)^{2g-2}\right). \tag{2.11}$$

The series (2.11) will be the basic object in this paper. It is the simplest and more natural formal power series in one variable obtained from topological string theory on a wide class of CY manifolds. As we have suggested, it corresponds to a perturbative gauge theory expansion in a conjectural large $N$ dual gauge theory. In some cases, the large $N$ dual can be identified explicitly. For example, if $X$ is the resolved conifold, the large $N$ dual is $U(N)$ Chern–Simons theory on $\mathbb{S}^3$ [41], and the formal power series (2.11) is the perturbative expansion of its partition function at fixed $N$.

We should note that, if we know the series $\Phi_{\boldsymbol{N}}(g_s)$ for all $r$-tuples of non-negative integers $\boldsymbol{N}$, we can reconstruct the power series $\mathcal{F}_g(\boldsymbol{\lambda})$ for all $g \geq 0$. Therefore, the collection of asymptotic series $\Phi_{\boldsymbol{N}}(g_s)$ contain the all-genus information of perturbative, topological string theory.

Another important motivation to consider the series (2.11) is the TS/ST correspondence formulated in [18, 19]. This correspondence provides a spectral theory interpretation for $\Phi_{\boldsymbol{N}}(g_s)$, as well as a matrix model interpretation [42, 43]. This will be relevant in concrete calculations of the integer invariants defined in this section.

## 2.2 Resurgence and Stokes constants

Formal power series with factorial growth lead, under some assumptions, to a universal mathematical structure described by the theory of resurgence, see [44, 45] for a formal description and [17, 46] for applications in gauge and string theory. This mathematical structure involves a family of numerical data called Stokes constants. When the perturbative series under consideration are topological invariants of some underlying geometric object, the corresponding Stokes constants lead to numerical, topological invariants. Surprisingly, these invariants turn out to be *integers* in many situations. In this section we will provide the formal definition of Stokes constants, and we will then apply this definition to the perturbative series (2.11).

---

[2]Note that we choose a slightly unusual convention for $g_s$ with an additional minus sign, for reasons explained in footnote 4.

Let us consider a factorially divergent (or Gevrey-1) series, of the form,

$$\varphi(\tau) = \sum_{n \geq 0} a_n \tau^n, \qquad a_n \sim n!. \tag{2.12}$$

The theory of resurgence tells us that, generically, this series will not be on its own, but will be accompanied by an additional set of formal power series. To understand how this comes about, we consider the Borel transform of $\varphi(\tau)$, defined by

$$\widehat{\varphi}(\zeta) = \sum_{k \geq 0} \frac{a_k}{k!} \zeta^k. \tag{2.13}$$

It is a holomorphic function in a neighborhood of the origin. In favorable cases, this function can be extended to the complex $\zeta$-plane (also called Borel plane), but it will have singularities. Let us label by an index $\omega$ the (perhaps infinite) set of singularities in the Borel plane $\zeta_\omega \in \mathbb{C}$. Let us assume for the moment being that this singularity is a logarithmic branch cut, and consider the behavior of $\widehat{\varphi}(\zeta)$ near $\zeta_\omega$:

$$\widehat{\varphi}(\zeta) = -S_\omega \frac{\log(\xi)}{2\pi i} \widehat{\varphi}_\omega(\xi) + \cdots \tag{2.14}$$

In this expression, $\xi = \zeta - \zeta_\omega$, the dots denote regular terms in $\xi$, the function

$$\widehat{\varphi}_\omega(\xi) = \sum_{n \geq 0} \hat{a}_{n,\omega} \xi^n \tag{2.15}$$

is an analytic function at $\xi = 0$, and $S_\omega$ will be later identified as Stokes constants. As the notation suggests, we can regard this series as the Borel transform of the Gevrey-1 series

$$\varphi_\omega(z) = \sum_{n \geq 0} a_{n,\omega} z^n, \qquad a_{n,\omega} = n! \hat{a}_{n,\omega}. \tag{2.16}$$

We then see that, starting with a single formal power series, we can obtain a new family of them by just looking at the behavior near the singularities of the Borel transform. We can now repeat the procedure with the series obtained in this way. Eventually, one ends up with a family of formal power series

$$\mathfrak{B}_\varphi = \{\varphi_\omega(z)\}_{\omega \in \Omega}, \tag{2.17}$$

which we will call the *minimal resurgent structure* associated to the original series $\varphi(z)$. The Borel transform of the formal power series in the minimal resurgent structure have local expansions

$$\widehat{\varphi}_\omega(\zeta) = -S_{\omega\omega'} \frac{\log(\xi)}{2\pi i} \widehat{\varphi}_{\omega'}(\xi) + \cdots. \tag{2.18}$$

The complex numbers $S_{\omega\omega'}$ are called *Stokes constants*. Note that their values depend on a choice of normalization of the series $\varphi_\omega(z)$.

The Stokes constants appear in a different type of calculation, which often is easier to perform in practice. Let $\varphi(z)$ be a generic Gevrey-1 series, and let us assume that the analytically continued Borel transform $\widehat{\varphi}(\zeta)$ does not grow too fast at infinity. The *Borel resummation* of $\varphi(z)$ is defined as the Laplace transform

$$s(\varphi)(z) = \int_0^\infty e^{-\zeta} \widehat{\varphi}(\zeta z) d\zeta. \tag{2.19}$$

This function, when it exists, is locally analytic, but has discontinuities at special rays in the $z$ plane. Let $\zeta_\omega$ be a singularity of $\widehat{\varphi}(\zeta)$. A ray in the Borel plane which starts at the origin and passes through $\zeta_\omega$ is called a *Stokes ray*. It is of the form $e^{i\theta} \mathbb{R}_+$, where

$$\theta = \arg(\zeta_\omega). \tag{2.20}$$

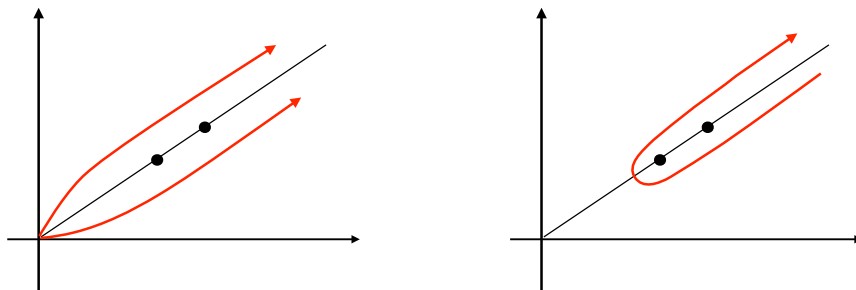

Figure 1: The difference between lateral Borel resummations in (2.23) (left) can be computed as a contour integral, given by the contribution of the singularities (right).

The integral (2.19) has a discontinuity whenever $\arg(z) = \arg(\zeta_\omega)$.

To calculate the discontinuities, we define the lateral Borel resummations for $z$ near a Stokes ray by

$$s_\pm(\varphi)(z) = \int_0^{e^{\pm i\epsilon}\infty} e^{-\zeta}\widehat{\varphi}(\zeta z)\mathrm{d}\zeta. \tag{2.21}$$

The discontinuity is then simply given by

$$\mathrm{disc}_\theta\varphi(z) = s_+(\varphi)(z) - s_-(\varphi)(z), \tag{2.22}$$

where $\theta$ is defined in (2.20). A simple contour-deformation calculation (see Fig. 1) shows that, in the case that all singularities are logarithmic, one has

$$s_+(\varphi_\omega)(z) - s_-(\varphi_\omega)(z) = \sum_{\omega'} \mathsf{S}_{\omega\omega'} e^{-\zeta_{\omega'}/z} s_-(\varphi_{\omega'})(z), \tag{2.23}$$

where the sum over $\omega'$ runs over all the singularities located on the Stokes ray. This equation turns out to give a powerful method to compute Stokes constants, by comparing (lateral) Borel resummations.

To write the fundamental equation (2.23) in a more convenient form, we note that the $\zeta_{\omega'}$ appearing in the r.h.s. depend implicitly on $\omega$. To have a more symmetric form, we introduce the *basic trans-series* $\Phi_\omega(z)$ as

$$\Phi_\omega(z) = e^{-V_\omega/z}\varphi_\omega(z), \tag{2.24}$$

where the $V_\omega$ are chosen in such a way that $\zeta_{\omega'} = V_{\omega'} - V_\omega$. The Borel resummation of the basic trans-series $\Phi_\omega(z)$ is defined by

$$s(\Phi_\omega)(z) = e^{-V_\omega/z}s(\varphi_\omega)(z). \tag{2.25}$$

To measure the discontinuity of Borel resummations across a Stokes ray, it is also useful to introduce the *Stokes automorphism* $\mathfrak{S}$ as

$$s_+ = s_-\mathfrak{S}. \tag{2.26}$$

In this way, (2.23) can be written as

$$\mathfrak{S}(\Phi_\omega) = \Phi_\omega + \sum_{\omega'} \mathsf{S}_{\omega\omega'}\Phi_{\omega'}. \tag{2.27}$$

Let us emphasize that the definition of Stokes constants given above relies on relatively mild assumptions. In technical terms, we are assuming that the formal power series $\varphi_\omega(z)$ are simple resurgent functions. The property of being resurgent means that, on every finite line issuing from the origin, the Borel transform $\widehat{\varphi}_\omega(\zeta)$ has a finite number of singularities, and one can analytically continue it along a path which circumvents these singularities. Simple resurgent functions have only logarithmic singularities, as in (2.14). However, it is straightforward to extend this formalism to resurgent functions with more general singularities, like branch cuts of the form

$$\widehat{\varphi}(\zeta_\omega + \xi) = (-\xi)^{-\nu} \frac{\mathsf{S}_\omega}{2\mathrm{i}\sin(\pi b)} \sum_{n\geq 0} \widehat{a}_n \xi^n + \text{regular}, \qquad \nu \notin \mathbb{Z}. \tag{2.28}$$

In this case, we regard $\widehat{\varphi}_\omega(\xi)$ as the Borel transform of

$$\varphi_\omega(z) = z^{-\nu} \sum_{n\geq 0} a_n z^n, \qquad a_n = \Gamma(n+1-\nu)\widehat{a}_n, \tag{2.29}$$

and it is easy to check that the discontinuity formula (2.23) still holds in the same form.

In summary, the machinery of resurgence gives a well-defined procedure to obtain a (possibly infinite) set of Stokes constants: starting from a single perturbative series $\varphi$, one first finds the minimal resurgent structure $\mathfrak{B}_\varphi$, and then the set of Stokes constants. Schematically, we have

$$\varphi(\tau) \longrightarrow \mathfrak{B}_\varphi = \{\varphi_\omega(\tau)\}_{\omega\in\Omega} \longrightarrow \{\mathsf{S}_{\omega\omega'}\}_{\omega,\omega'\in\Omega}. \tag{2.30}$$

**Remark 2.1.** At this point, it might be useful to add some clarifying remarks on the notion of minimal resurgent structure introduced above. The minimal resurgent structure associated to a formal series $\varphi(z)$ is the smallest set of resurgent functions which is needed to find a closed system under Stokes automorphisms. It does not include necessarily all the formal power series in the theory. A surprising example of this situation occurs in complex CS theory, where formal power series are associated to perturbative expansions around flat connections on a three-manifold $M$. If $M$ is the complement of a hyperbolic knot, there are two special flat connections, namely the trivial connection, and the so-called geometric connection, which corresponds to the unique complete hyperbolic metric on $M$. They lead to two different formal power series, which we denote by $\Phi_0$ and $\Phi_g$, respectively. It follows e.g. from the results in [11,12,47,48] that the minimal resurgent structure associated to $\Phi_g$, $\mathfrak{B}_{\Phi_g}$, does *not* include $\Phi_0$. The minimal resurgent structure associated to $\Phi_0$ is however strictly bigger: $\mathfrak{B}_{\Phi_g} \subsetneq \mathfrak{B}_{\Phi_0}$ [48–50]. □

## 2.3 Stokes constants in topological string theory

The Stokes constants defined above can be finite or infinite in number, depending on the cardinality of the set $\Omega$ which labels the basic trans-series of the model. They are, in general, complex numbers. In the case of asymptotic series associated to irregular singularities of nonlinear ODEs, like the Painlevé equations, there is an infinite number of Stokes constants, and they are transcendental numbers. In many examples of linear ODEs, the Stokes constants are a finite set of integers. However, there is a third category of examples involving *infinitely many integer Stokes constants*. This is the case of "peacock patterns," which was studied in detail in [11, 12] in the case of complex CS theory. We conjecture that peacock patterns are typical of theories based on quantum curves. This includes topological string theory on toric CY threefolds (as we discuss in this paper), (complex) CS theory, and the exact WKB method for generic difference equations.

Let us now discuss in more detail how to apply the framework of the previous section to the perturbative series defined in (2.11). We first recall that, as explained in (2.11), for every set

of integers $N$, topological string theory provides a *single* perturbative series $\Phi_N(g_s)$. It is easy to verify in examples that these series are Gevrey-1 and resurgent functions, and we conjecture that this is the case in general. We also expect that each of these series will be accompanied by additional series (or basic trans-series), leading to a minimal resurgent structure $\mathfrak{B}_{\Phi_N}$. This structure can be found by studying the singularities of their Borel transforms.

This is indeed the case, and, as we will see in the examples discussed below, one finds a "peacock pattern structure" very similar to the one described in [11,12] in complex CS theory. More precisely, the minimal resurgent structure $\mathfrak{B}_{\Phi_N}$ consists of the following ingredients. There is a *finite* number of Gevrey-1 series

$$\Phi_{\sigma;N}(g_s), \qquad \sigma = 0, 1, \cdots, \ell_N, \tag{2.31}$$

where $\Phi_{0,N}(g_s) = \Phi_N(g_s)$ is the original series (2.11). The total number of these series, $\ell_N + 1$, depends, as we have indicated, on $N$, and on the CY we are considering. In addition to this finite number of basic trans-series, we have an infinite family of them, labelled by an additional integer $n \in \mathbb{Z}$, and of the form

$$\Phi_{\sigma,n;N}(g_s) = \Phi_{\sigma;N}(g_s)e^{-n\frac{\mathcal{A}}{g_s}}, \qquad n \in \mathbb{Z}, \tag{2.32}$$

where $\mathcal{A}$ is a complex constant which depends on $N$ and on the CY. Note in particular that the singularities of the Borel transforms $\widehat{\Phi}_{\sigma;N}(\zeta)$ form infinite towers in the Borel plane, spaced by integer multiples of $\mathcal{A}$. The minimal resurgent structure is then of the form

$$\mathfrak{B}_{\Phi_N} = \{\Phi_{\sigma,n;N}(g_s)\}_{\sigma=0,\cdots,\ell_N;\, n\in\mathbb{Z}}. \tag{2.33}$$

Given the structure (2.32), the Stokes constants $\mathsf{S}_{\sigma\sigma',n;N}$ are labelled by a pair of indices $\sigma, \sigma' = 0, \cdots, \ell_N$ and an integer $n$, and we conjecture that they are *integers*, after a natural normalization of the fundamental series in (2.31). They can be naturally organized as $q$-series,

$$\mathsf{S}_{\sigma\sigma';N}(q) = \sum_{n\in\mathbb{Z}} \mathsf{S}_{\sigma\sigma',n;N} q^n. \tag{2.34}$$

Let us note that the formal series (2.32) represent additional sectors of the topological string which are invisible in perturbation theory (in particular, they involve explicit non-perturbative corrections in the string coupling constant $g_s$). We believe that they are closely related to the trans-series constructed in [29,30] by using the holomorphic anomaly. At this moment, however, the geometric and physical meaning of these sectors is not clear.

The above procedure can be summarized as a "resurgent machine" in which we start with perturbative series and we end up, conjecturally, with integer invariants organized in $q$ series. Schematically,

$$\text{perturbative series} \longrightarrow \text{integer invariants}/q\text{-series}. \tag{2.35}$$

We should mention that this procedure to obtain integer invariants and $q$-series from perturbative series is different from the one suggested in [51–57], in the context of Chern–Simons theory with a compact gauge group. In those papers, the perturbative series is upgraded to a $q$-series, in such a way that the former is recovered through the so-called radial asymptotics of the latter. In that upgrading, the $q$-series is not defined in a unique way from the perturbative series. In contrast, in the procedure described above, the perturbative series determines uniquely the different $q$-series, since it determines uniquely the set of Stokes constants.

# 3 Examples

We will now present some non-trivial examples of the above construction, focusing on the case of toric CY manifolds[3]. A very useful tool in the determination of the Stokes constants defined in the previous section is the TS/ST correspondence, which we now summarize. A more detailed review and references can be found in [58].

## 3.1 The TS/ST correspondence

The simplest case of the TS/ST correspondence corresponds to the case of toric CYs $X$ with one single true modulus, and we will only consider this type of examples in this paper. In this case, the mirror curve $\Sigma_X$ has genus one. As shown in [18], there is a canonical way to obtain an operator on $L^2(\mathbb{R})$, $\rho_X$, by Weyl quantization of this curve. The mass parameters of $X$ become parameters of the operator $\rho_X$. The quantization procedure introduces a Planck constant $\hbar \in \mathbb{R}_{>0}$. It can be shown that, for appropriate values of the mass parameters, the operator $\rho_X$ is self-adjoint and of trace class, therefore its spectral determinant

$$\Xi_X(\kappa, \hbar) = \det(1 + \kappa \rho_X) \tag{3.1}$$

exists and defines an entire function of $\kappa$. Its expansion around the origin

$$\Xi_X(\kappa, \hbar) = 1 + \sum_{N \geq 1} Z_N(\hbar) \kappa^N \tag{3.2}$$

defines "fermionic spectral traces" $Z_N(\hbar)$, which depend on $\hbar$ and on the possible mass parameters of $X$. It is easy to show that these fermionic traces are polynomials in the conventional "bosonic" traces

$$\operatorname{Tr} \rho_X^n. \tag{3.3}$$

In the more general case in which the toric CY has $r$ true moduli, one can define a generalized spectral determinant depending on $r$ variables $\boldsymbol{\kappa} = (\kappa_1, \cdots, \kappa_r)$, with an expansion of the form [19,39]

$$\Xi_X(\boldsymbol{\kappa}, \hbar) = 1 + \sum_{\boldsymbol{N}} Z_{\boldsymbol{N}}(\hbar) \kappa_1^{N_1} \cdots \kappa_r^{N_r}. \tag{3.4}$$

The TS/ST correspondence postulates, among other things, that the fermionic spectral traces $Z_N(\hbar)$ have an asymptotic expansion as $\hbar \to \infty$ which is essentially given by the formal power series (2.11) [4]

$$Z_N(\hbar) \sim c_N g_s^{\gamma_N} \Phi_N(g_s), \qquad \hbar = -g_s^{-1} \gg 1. \tag{3.5}$$

Here, $c_N$ and $\gamma_N$ are constants. Since the fermionic spectral traces are well-defined for $\hbar \in \mathbb{R}_{>0}$, this defines a rigorous non-perturbative completion of topological string theory at the conifold point. This particular consequence of the TS/ST correspondence has been tested in detail and extensively studied in e.g. [19,39,42,43,59,60]. It should be noted that the asymptotic conjecture (3.5) is itself the consequence of a stronger conjecture which provides an *exact* expression for the spectral determinant and the fermionic spectral traces in terms of topological string data (and, in particular, of BPS invariants of the CY $X$). One can obtain in this way exact

---

[3]The simplest toric CY manifold is the resolved conifold. However, in that case the series $\Phi_N(g_s)$ has a finite radius of convergence, there are no singularities in the Borel plane, and the Stokes constants vanish. Non-trivial Stokes constants only arise when the toric CY has compact four-cycles.

[4]We choose this convention for $g_s$ because we want to treat $\hbar$ and $g_s$ as modular-like parameters, and the relation $\hbar = -g_s^{-1}$ is the analogue of an $S$-type modular transformation.

quantization conditions for the spectral problem of the quantum mirror curve (see also [61–64] for related developments).

As we will show, the relation (3.5) turns out to be very useful to study the resurgent structure of the formal power series $\Phi_N(g_s)$. However, to understand this structure, we have to consider the case in which $g_s$ (or, equivalently, $\hbar$) is *complex*. The complexification of $\hbar$ in the TS/ST correspondence was addressed in different forms in [65–68]. In [66, 67] it was shown in some examples that the spectral problem associated to the operator $\rho_X$ is well-defined when $\hbar$ is of the form $\hbar = 2\pi e^{i\theta}$, where $\theta \in (0, \pi)$. The spectrum of the operator $\rho_X$ is in this case discrete and complex.

In this paper we will assume that TS/ST correspondence can be extended to the complex $\hbar$-plane minus the negative real axis,

$$\hbar \in \mathbb{C}' = \mathbb{C}\backslash\mathbb{R}_{\leq 0}, \tag{3.6}$$

in such a way that the relevant operators remain of trace class, and the fermionic spectral traces $Z_N(\hbar)$ are *analytic functions* on $\mathbb{C}'$. Although we do not know how to establish this on general grounds, we will show that it is possible to perform the analytic continuation to $\mathbb{C}'$ in explicit expressions for the $Z_N(\hbar)$. It turns out that the resulting picture shares many formal similarities with complex Chern–Simons theory on the complement of a hyperbolic knot $\mathcal{K}$. More precisely, we have the following parallelisms:

1. The spectral determinant $\Xi_X(\kappa, \hbar)$ and the fermionic spectral traces $Z_N(\hbar)$ correspond to state integral invariants of the knot [26, 27], also known as Andersen–Kashaev invariants [28]. For the complement of a hyperbolic knot, the Andersen–Kashaev invariant is a function $Z_{\mathcal{K}}(u, \tau)$ of the holonomy $u$ around the knot, and a complex coupling constant $\tau$ which plays the rôle of Planck's constant. Therefore, we expect that the full spectral determinant $\Xi_X(\kappa, \hbar)$, as function of the modulus $\kappa$, corresponds to the state integral as a function of $u$. Note that both are entire functions of the moduli (for $Z_{\mathcal{K}}(u, \tau)$, this was proved in [12] in the case of the first two non-trivial hyperbolic knots). Accordingly, the spectral traces $Z_N(\hbar)$ correspond to the coefficients of $Z_{\mathcal{K}}(u, \tau)$ in a Taylor expansion around $u = 0$. The first non-trivial coefficient in this expansion, which is $Z_{\mathcal{K}}(\tau) = Z_{\mathcal{K}}(u = 0, \tau)$, was studied in [11] (the case with more than one modulus in topological string theory would correspond to the case of links in complex CS theory).

2. The asymptotic expansion (3.5) of $Z_N(\hbar)$, involving the perturbative topological string series, corresponds to the expansion of the Andersen–Kashaev invariant at large positive values of $\tau$. This is given by the asymptotic expansion of complex CS theory around the so-called *conjugate* connection (which is the complex conjugate to the geometric connection).

3. The *volume conjecture* for the Andersen–Kashaev invariant [28] states that its asymptotics at $\tau \to 0^+$ involves the complexified volume[5] $V_{\mathcal{K}}$ of the three sphere complement of the hyperbolic knot $\mathcal{K}$:

$$Z_{\mathcal{K}}(\tau) \sim \exp\left(-\frac{V_{\mathcal{K}}}{\tau}\right), \qquad \tau \to 0^+. \tag{3.7}$$

In this equation, we have set $u = 0$ in $Z_{\mathcal{K}}(u, \tau)$, but there is a generalization to arbitrary $u$ [28, 69]. The conjecture (3.7), which is closely related to the original Kashaev conjecture [70], relates a "quantum" invariant of the knot in the l.h.s., to a "classical" geometric invariant in the r.h.s. Interestingly, the TS/ST correspondence implies a *conifold volume*

---

[5]The imaginary part of the complexified volume is the Chern-Simons action.

Table 1: Similarities between structures appearing in complex CS theory and in the TS/ST correspondence.

| complex CS theory | TS/ST correspondence |
|---|---|
| state integral | spectral determinant and traces |
| expansion around conjugate connection | perturbative topological string |
| volume conjecture | conifold volume conjecture |
| (symmetric) factorization | (non-symmetric) factorization |
| DGG index | new integer invariants |

*conjecture for toric CYs*,

$$Z_N(-g_s^{-1}) \sim \exp\left( \frac{1}{g_s} \sum_{i=1}^r N_i \mathcal{V}_i \right), \qquad g_s \to 0^-, \qquad (3.8)$$

where $\mathcal{V}_i$, $i = 1, \cdots, r$ are the values of the Kähler parameters of the CY at the conifold point. This can be also regarded as a relation between the asymptotics of a "quantum" invariant on the l.h.s. (a spectral trace obtained by quantizing the mirror curve), to a "classical" geometric property of the CY. Let us note as well that the $\mathcal{V}_i$ can be regarded as the "minimal" values of the volumes of the CY manifold. The conifold volume conjecture (3.8) for toric CY manifolds was checked in one and two-moduli examples in [19, 39, 42, 43, 71].

4. The state integral has a factorization property, namely, it can be written as a sum of products of holomorphic and anti-holomorphic blocks [11, 12, 72–75]. Moreover, the blocks are given by $q$ and $\tilde{q}$-series, where

$$q = e^{2\pi i \tau}, \qquad \tilde{q} = e^{-2\pi i/\tau}. \qquad (3.9)$$

As we will see, the fermionic spectral traces also factorize in various examples, and can be written as a sum of products of $q$ and $\tilde{q}$ series. There is however an important difference between the state integral and the fermionic spectral traces. The former is symmetric under the $S$-duality transform

$$\tau \to -\frac{1}{\tau} \qquad (3.10)$$

which exchanges $q$ and $\tilde{q}$. This leads to a symmetry between the holomorphic and anti-holomorphic blocks, which involve the same functions (with different arguments). In contrast, the fermionic spectral traces do *not* have this symmetry, and sending $\hbar$ to $-1/\hbar$ exchanges the conventional topological string with the NS topological string. As a consequence, the holomorphic and anti-holomorphic blocks are given by different functions, as we will see in examples.

5. As shown in [11, 12], the Stokes constants associated to the different asymptotic series in complex CS theory are integer invariants, and they turn out to be closely related to the DGG index of the knot [3]. In the case of the topological string, the Stokes constants are precisely the new integer invariants that we define in this paper, therefore we can regard these as analogues of the DGG index.

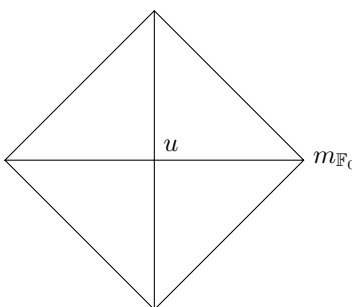

Figure 2: The toric diagram of the local $\mathbb{F}_0$ geometry.

We summarize these parallelisms in Table 1. The formal similarities between the two theories suggest a reinterpretation of topological string theory on toric CYs in terms of a topological gauge theory, or perhaps of a superconformal field theory in 3d, in view of the 3d/3d correspondence [31,76][6]. We should note that, even at the formal level, the topological string story seems to be much more involved than the complex CS counterpart. For example, in complex CS, the different formal power series have a simple semiclassical interpretation in terms of expansions around different saddle points, and this makes their calculation a relatively easy task. We do not have such a physical interpretation for the topological string series (2.31). Another crucial difference is that the full moduli-dependent state integral $Z_{\mathcal{K}}(u, \tau)$ can be represented in terms of an integral, which can be in addition calculated explicitly in many interesting examples. In the case of the topological string, we are not aware of integral expressions for the spectral determinant.

As a final comment on the formal similarities between CS and topological string theory, let us point out the following. One could think that $Z_{\mathcal{K}}(u, \tau)$, being a wavefunction, corresponds rather to an open topological string theory. However, as noted in [58], it is likely that $\Xi(\kappa, \hbar)$ has an interpretation as a wavefunction on the moduli space of the closed CY moduli, in line with the suggestion of [77]. One important open problem is to find the analogue of the quantum differential equation satisfied by $\Xi(\kappa, \hbar)$, similar to the AJ equation satisfied by $Z_{\mathcal{K}}(u, \tau)$. Inspired by results in the 4d limit of the theory [78], it has been suggested that $q$-Painlevé equations could play such a rôle [79], but a complete description is still lacking. Recently, and following the same philosophy, [80] obtained a "wave equation" in the case of the resolved conifold. Connections between partition functions in the 4d case and integrable equations have been also studied in [81–83].

## 3.2 Local $\mathbb{F}_0$

The first example we consider is the canonical bundle over the Hirzebruch surface $\mathbb{F}_0 = \mathbb{P}^1 \times \mathbb{P}^1$, which is also called the local $\mathbb{F}_0$ geometry,

$$X = \mathcal{O}(-2, -2) \to \mathbb{P}^1 \times \mathbb{P}^1. \tag{3.11}$$

This geometry is a toric variety and it is described by the toric diagram in Fig. 2. The Kähler moduli of this geometry are one true modulus $u$ corresponding to the internal vertex of the toric diagram, and one mass parameter $m_{\mathbb{F}_0}$ corresponding to one of the four boundary vertices. For the topological string theory on local $\mathbb{F}_0$, the full moduli space is identified with the family of mirror curves described by the equation

$$e^x + m_{\mathbb{F}_0} e^{-x} + e^y + e^{-y} + \tilde{u} = 0, \quad x, y \in \mathbb{C}. \tag{3.12}$$

---

[6]The possible relationship with a 3d superconformal field theory was already pointed out in [18], where it was also noted that the spectral traces at large $N$ and fixed $\hbar$ satisfy a $N^{3/2}$ scaling typical of many 3d theories.

The moduli are a "true" modulus $u = 1/\tilde{u}^2$ and the mass parameter $m_{\mathbb{F}_0}$. The maximal conifold point on the moduli space is in the discriminant of (3.12). For simplicity we only consider $m_{\mathbb{F}_0} = 1$, in which case the maximal conifold point is located at

$$u = \frac{1}{16}. \tag{3.13}$$

The free energies of the topological string at the maximal conifold point can be computed by the holomorphic anomaly equations [84,85], and in terms of the local flat coordinate $\lambda$ they are [43,86]

$$\mathcal{F}_0(\lambda, m_{\mathbb{F}_0} = 1) = \frac{\lambda^2}{2}\left(\log\lambda + \log\left(\frac{\pi^2}{4}\right) - \frac{3}{2}\right) - \frac{2C}{\pi^2} - \frac{\pi^2\lambda^3}{12} + \frac{5\pi^4\lambda^4}{288} - \frac{7\pi^6\lambda^5}{960} + \ldots, \tag{3.14a}$$

$$\mathcal{F}_1(\lambda, m_{\mathbb{F}_0} = 1) = -\frac{1}{12}\log(\lambda) + \frac{\pi^2\lambda}{24} + \frac{13\pi^4\lambda^2}{288} - \frac{29\pi^6\lambda^3}{576} + \ldots, \tag{3.14b}$$

$$\mathcal{F}_2(\lambda, m_{\mathbb{F}_0} = 1) = -\frac{1}{240\lambda^2} + \frac{53\pi^6\lambda}{1920} + \ldots, \tag{3.14c}$$

where $C = \text{Im}(\text{Li}_2(i)) = 0.915966\ldots$ is the Catalan's constant, and we have restricted the mass parameter $m_{\mathbb{F}_0} = 1$. We can then construct the formal power series $\Phi_{0;N}(g_s)$ by (2.11).

In this section we are only concerned with the case of $N = 1$. We will see that this example already has a very rich resurgent structure, and it is very similar to the quantum invariants of the figure eight knot studied in [11,12]. Furthermore, we will show the power of the TS/ST correspondence, which enables us to write down the *complete* set of Stokes constants.

### 3.2.1 The first resurgent structure

By using the free energies (3.14a)–(3.14c) we can construct the first asymptotic series for $N = 1$ by (2.11)

$$\Phi_{0;1}(g_s) = e^{\mathcal{V}_{\mathbb{F}_0}/g_s}\varphi_{0;1}(g_s), \tag{3.15}$$

where

$$\mathcal{V}_{\mathbb{F}_0} = \frac{2C}{\pi^2}, \tag{3.16}$$

and

$$\varphi_{0;1}(g_s) = 1 + \frac{\pi^2}{24}g_s + \frac{73\pi^4}{1152}g_s^2 + \frac{13541\pi^6 g_s^3}{414720} + \frac{855509\pi^8 g_s^4}{39813120} + \frac{150067879\pi^{10} g_s^5}{6688604160} + \ldots. \tag{3.17}$$

Since we are only concerned with $N = 1$, we will suppress the subscript $_{;1}$ in the remainder of the section.

The series $\Phi_0(g_s)$ has a very rich resurgent structure. The Borel transform $\widehat{\Phi}_0(\zeta)$ has two infinite towers of singularities located at

$$in, \quad n \in \mathbb{Z}_{\neq 0}, \tag{3.18}$$

$$\zeta_0 + in', \quad n' \in \mathbb{Z}, \tag{3.19}$$

with $\zeta_0 = 2\mathcal{V}_{\mathbb{F}_0}$, which is illustrated in the left panel of Fig. 3. The first tower of singularities along the imaginary axis corresponds to the following asymptotic series in the same family

$$\Phi_{0,n}(g_s) = \Phi_0(g_s)e^{-n\frac{i}{g_s}}, \quad n \in \mathbb{Z}. \tag{3.20}$$

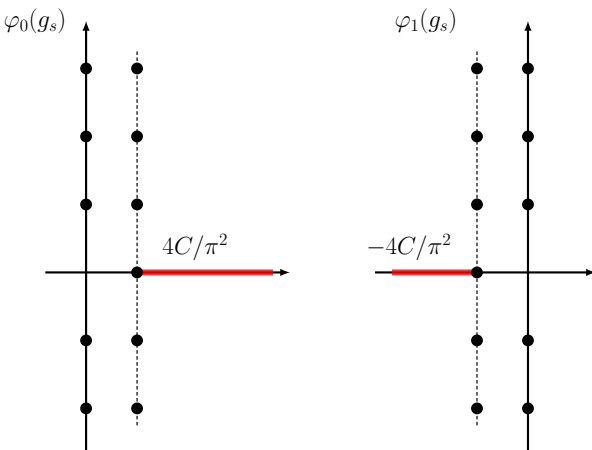

Figure 3: The singularities in the Borel plane for the series $\varphi_0(g_s)$, $\varphi_1(g_s)$.

The second tower of singularities indicates the existence of a new family of asymptotic series. Since the singularity $\zeta_0$ on the positive real axis is the nearest singularity, it controls the large order behavior of the coefficients $a_n$ of $\varphi_0(g_s)$ by

$$
\begin{aligned}
a_n &\sim \frac{\mathsf{S}_{01,0}}{2\pi\mathrm{i}} \sum_{k=0}^{\infty} b_k \zeta_0^{k-n} \Gamma(n-k) \\
&= \frac{\mathsf{S}_{01,0}}{2\pi\mathrm{i}} \zeta_0^{-n} \Gamma(n) \left( b_0 + \frac{b_1 \zeta_0}{n-1} + \frac{b_2 \zeta_0^2}{(n-1)(n-2)} + \dots \right),
\end{aligned}
\tag{3.21}
$$

where $b_k$ are the coefficients of the series that resurges at $\zeta_0$, and $\mathsf{S}_{01,0}$ is the Stokes constant. By systematically extracting the coefficients $b_k$, we find that the second family of series is given by

$$
\Phi_{1,n}(g_s) = \Phi_1(g_s) \mathrm{e}^{-n\frac{\mathrm{i}}{g_s}}, \quad n \in \mathbb{Z},
\tag{3.22}
$$

with

$$
\Phi_1(g_s) = \mathrm{i}\Phi_0(-g_s).
\tag{3.23}
$$

Along the way, we also find the value of the first Stokes constant

$$
\mathsf{S}_{01,0} = 4.
\tag{3.24}
$$

Next we study the resurgent structure of the new series $\Phi_1(g_s)$. Since it is related to $\Phi_0(g_s)$ by (3.23), its Borel singularities are mirror reflection of those of $\Phi_0(g_s)$ with respect to the imaginary axis, as illustrated in the right panel of Fig. 3. They are located at

$$
\mathrm{i}n, \quad n \in \mathbb{Z}_{\neq 0},
\tag{3.25}
$$

$$
\zeta_1 + \mathrm{i}n', \quad n' \in \mathbb{Z},
\tag{3.26}
$$

with $\zeta_1 = -4C/\pi^2$, associated with $\Phi_{1,n}(g_s)$ and $\Phi_{0,n'}(g_s)$ respectively. In addition, using the same large order analysis we find that the Stokes constant at the singularity $\zeta_1$ on the negative real axis is

$$
\mathsf{S}_{10,0} = -4.
\tag{3.27}
$$

Therefore we conclude that $\{\Phi_{0,n}(g_s), \Phi_{1,n}(g_s)\}$ form a minimal resurgent structure. We denote by $\Phi(g_s)$ the 2-vector of asymptotic series

$$
\Phi(g_s) = \begin{pmatrix} \Phi_0(g_s) \\ \Phi_1(g_s) \end{pmatrix}.
\tag{3.28}
$$

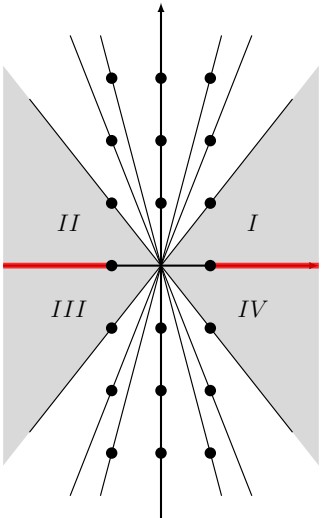

Figure 4: Stokes rays and sectors in the $g_s$-plane for $\Phi(g_s)$.

As shown in Fig. 4, the Stokes rays passing through the Borel singularities of both series arrange into a peacock pattern, very similar to the one found for the asymptotic series of the figure eight knot in [11, 12]. The Stokes rays divide the complex plane into infinitely many sectors, and only inside a sector can the Borel resummation of $\Phi(g_s)$ be defined. We denote by $s_R(\Phi)(g_s)$ the Borel resummation along a ray in the sector $R$ of the asymptotic series. As in [11, 12] we denote the four sectors bordering the real axis by $I, II, III, IV$ in the anti-clockwise order.

Following the structure of Borel singularities, we can collect all the Stokes constants into the matrix

$$\mathsf{S}(q) = (\mathsf{S}_{\sigma\sigma'}(q))_{\sigma,\sigma'=0,1}. \tag{3.29}$$

In this section we will scale powers of $q$ by $1/2$ in the generating series of Stokes constants for the reason that will become clear in the next subsection, in other words

$$\mathsf{S}_{\sigma,\sigma'}(q) = \sum_{n \in \mathbb{Z}} \mathsf{S}_{\sigma\sigma',n} q^{n/2}. \tag{3.30}$$

We also decompose the matrix $\mathsf{S}(q)$ as

$$\mathsf{S}(q) = \mathsf{S}^{(0)} + \mathsf{S}^+(q) + \mathsf{S}^-(q), \tag{3.31}$$

where $\mathsf{S}^{(0)}$ is an off-diagonal constant matrix containing Stokes constants on the real axis, the entries of $\mathsf{S}^+(q)$ are $q$-series encoding Stokes constants in the upper half plane, while the entries of $\mathsf{S}^-(q)$ are $q^{-1}$-series encoding Stokes constants in the lower half plane.

The constant matrix $\mathsf{S}^{(0)}$ has only two non-vanishing off-diagonal entries

$$\mathsf{S}^{(0)} = \begin{pmatrix} 0 & \mathsf{S}_{01,0} \\ \mathsf{S}_{10,0} & 0 \end{pmatrix}, \tag{3.32}$$

and they were already obtained in (3.24), (3.27). One finds that the matrix $\mathsf{S}^{(0)}$ is skew-symmetric. To compute the Stokes matrices $\mathsf{S}^+(q)$ and $\mathsf{S}^-(q)$ it is beneficial to use the TS/ST correspondence as well as more advanced machinery of radial asymptotic analysis, which we will explain in detail in the following two subsections. We quote the results here for completeness. The Stokes automorphism $\mathfrak{S}_{I \to II}(q)$ from sector $I$ to $II$ is defined by

$$s_{II}(\Phi)(g_s) = \mathfrak{S}_{I \to II}(q) s_I(\Phi)(g_s), \tag{3.33}$$

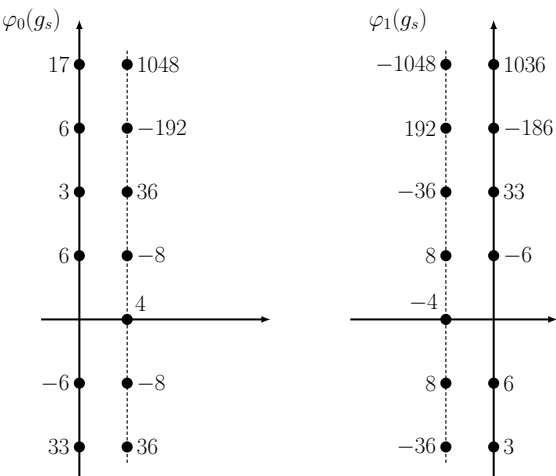

Figure 5: Stokes constants for the minimal resurgent structure $\{\Phi_{0,n}(g_s),\Phi_{1,n}(g_s)\}$ of local $\mathbb{F}_0$.

and its explicit expression is

$$\mathfrak{S}_{I\mapsto II}(q) = \frac{1}{2}\begin{pmatrix} 2g(q)G(q) & G'(q)g(q)-g'(q)G(q) \\ G(q)g'(q)-g(q)G'(q) & 2g'(q)G'(q), \end{pmatrix}, \tag{3.34}$$

where the $q$-series $g(q), G(q), g'(q), G'(q)$ are defined in (3.75), (3.77a), (3.77b). We note that $\mathfrak{S}_{I\mapsto II}(q)$ is also skew-symmetric. The Stokes constants in the upper half plane can be read off from $\mathfrak{S}_{I\mapsto II}(q)$. In particular,

$$\begin{aligned}
\mathsf{S}_{00}^+(q) &= \mathfrak{S}_{I\mapsto II}(q)_{1,1} - 1 \\
&= 6q^{1/2} + 3q + 6q^{3/2} + 17q^2 - 26q^{5/2} + 52q^3 + \dots, \tag{3.35a}
\end{aligned}$$

$$\begin{aligned}
\mathsf{S}_{01}^+(q) &= \mathfrak{S}_{I\mapsto II}(q)_{1,2}/\mathfrak{S}_{I\mapsto II}(q)_{1,1} \\
&= -8q^{1/2} + 36q - 192q^{3/2} + 1048q^2 - 5752q^{5/2} + 31656q^3 + \dots, \tag{3.35b}
\end{aligned}$$

$$\begin{aligned}
\mathsf{S}_{10}^+(q) &= \mathfrak{S}_{I\mapsto II}(q)_{2,1}/\mathfrak{S}_{I\mapsto II}(q)_{1,1} \\
&= 8q^{1/2} - 36q + 192q^{3/2} - 1048q^2 + 5752q^{5/2} - 31656q^3 + \dots, \tag{3.35c}
\end{aligned}$$

$$\begin{aligned}
\mathsf{S}_{11}^+(q) &= \mathfrak{S}_{I\mapsto II}(q)_{2,2} - \mathfrak{S}_{I\mapsto II}(q)_{1,2}\mathfrak{S}_{I\mapsto II}(q)_{1,1}\mathfrak{S}_{I\mapsto II}(q)_{2,1} - 1 \\
&= -6q^{1/2} + 33q + 582q^{3/2} + 1420q^2 + 2528q^{5/2} + 7383q^3 + \dots. \tag{3.35d}
\end{aligned}$$

Note that the definition of the Stokes automorphism (3.33), together with the difference between neighboring series in the same family (3.20), (3.22) implies that $q$ is identified naturally with

$$q = e^{-\frac{2i}{g_s}}. \tag{3.36}$$

To compute the Stokes constants in the lower half plane, we notice that

$$\Phi(g_s)^* = \begin{pmatrix} 1 & 0 \\ 0 & -1 \end{pmatrix}\Phi(g_s^*), \quad q(g_s)^* = q(g_s^*)^{-1}. \tag{3.37}$$

Therefore by complex conjugating both sides of (3.33) we find

$$s_{IV}(\Phi)(g_s) = \mathfrak{S}_{III\mapsto IV}(q)s_{III}(\Phi)(g_s), \tag{3.38}$$

with

$$\mathfrak{S}_{III\mapsto IV}(q) = \begin{pmatrix} 1 & 0 \\ 0 & -1 \end{pmatrix}\mathfrak{S}_{I\mapsto II}(q^{-1})^{-1}\begin{pmatrix} 1 & 0 \\ 0 & -1 \end{pmatrix}. \tag{3.39}$$

The Stokes constants in the lower half plane can also be read off and we find

$$S_{00}^{-}(q) = -6q^{-1/2} + 33q^{-1} + 582q^{-3/2} + 1420q^{-2} + \dots, \tag{3.40a}$$

$$S_{01}^{-}(q) = -8q^{-1/2} + 36q^{-1} - 192q^{-3/2} + 1048q^{-2} + \dots, \tag{3.40b}$$

$$S_{10}^{-}(q) = 8q^{-1/2} - 36q^{-1} + 192q^{-3/2} - 1048q^{-2} + \dots, \tag{3.40c}$$

$$S_{11}^{-}(q) = 6q^{-1/2} + 3q^{-1} + 6q^{-3/2} + 17q^{-2} + \dots. \tag{3.40d}$$

We illustrate the Stokes constants in Fig. 5.

### 3.2.2 Spectral trace and its factorisation

According to the TS/ST correspondence [18, 19], we should promote the mirror curve (3.12) to the difference operator

$$\mathcal{O}_{\mathbb{F}_0}(m_{\mathbb{F}_0}) = e^{\mathsf{x}} + m_{\mathbb{F}_0} e^{-\mathsf{x}} + e^{\mathsf{y}} + e^{-\mathsf{y}}, \tag{3.41}$$

where $\mathsf{x}$ and $\mathsf{y}$ are Heisenberg operators, satisfying the commutation relation

$$[\mathsf{x}, \mathsf{y}] = i\hbar. \tag{3.42}$$

Then the inverse operator $\rho_{\mathbb{F}_0} = \mathcal{O}_{\mathbb{F}_0}^{-1}$ is of trace class [59] and its Fredholm determinant as well as fermionic spectral traces $Z_N(\hbar)$ can be defined. Furthermore the latter can be computed explicitly [43].

The integral kernel for $\rho_{\mathbb{F}_0}$ was obtained in [43] for real $\hbar$, and it reads

$$\rho_{\mathbb{F}_0}(x_1, x_2) = \frac{e^{-b^2\xi/2 + \pi b(x_1+x_2)/2}}{2b\cosh(\pi\frac{x_1-x_2}{b})} \frac{\Phi_b(x_1 - b\xi/2\pi + ib/4)\,\Phi_b(x_2 + b\xi/2\pi + ib/4)}{\Phi_b(x_1 + b\xi/2\pi - ib/4)\,\Phi_b(x_2 - b\xi/2\pi - ib/4)}. \tag{3.43}$$

The parameter $\xi$ is related to the mass parameter $m_{\mathbb{F}_0}$ by

$$m_{\mathbb{F}_0} = e^{2b^2\xi}, \tag{3.44}$$

and b is related to $\hbar$ by

$$\hbar = \pi b^2. \tag{3.45}$$

As we mentioned above, we need to perform an analytic continuation of the spectral theory of $\rho_{\mathbb{F}_0}$ to complex $\hbar$. Since we have an explicit expression (3.43) for its integral kernel, this is easy to do. We just note that Faddeev's quantum dilogarithm can be analytic continued to all values of b such that $b^2 \notin \mathbb{R}_{\leq 0}$. Therefore, (3.43) defines the integral kernel of the operator for $\hbar \in \mathbb{C}'$, as required in (3.6).

The first spectral trace has the following integral representation:

$$\text{Tr}\rho_{\mathbb{F}_0} = \frac{e^{-b^2\xi/2}}{2b} \int_{\mathbb{R}} e^{\pi bx} \frac{\Phi_b(x - b\xi/2\pi + ib/4)\,\Phi_b(x + b\xi/2\pi + ib/4)}{\Phi_b(x + b\xi/2\pi - ib/4)\,\Phi_b(x - b\xi/2\pi - ib/4)} dx, \tag{3.46}$$

while the fermionic trace is

$$Z_1(m_{\mathbb{F}_0}, \hbar) = \text{Tr}\rho_{\mathbb{F}_0}. \tag{3.47}$$

We will only be interested in the simple case $\xi = 0$, where the integral reads

$$\text{Tr}\rho_{\mathbb{F}_0} = \frac{1}{2b} \int_{\mathbb{R}} e^{\pi bx} \frac{\Phi_b(x + ib/4)^2}{\Phi_b(x - ib/4)^2} dx. \tag{3.48}$$

Note that the integrand mostly consists of the product of quantum dilogarithms, and the spectral trace is therefore very similar to state integrals in complex Chern-Simons theory. Due to this reason, we will sometimes refer to the integral representation of spectral traces as state integrals.

The integrand of (3.48) is integrable for $\mathrm{Re}\, \mathrm{b} > 0$, which we always assume, and the spectral trace (3.48) is an analytic function of $\hbar \in \mathbb{C}'$. When $\hbar > 0$, a conjecture of [18] states that it can be computed in terms of the so-called modified grand potential of local $\mathbb{F}_0$, which is fully specified by BPS invariants of the CY threefold. This conjecture can be extended to complex $\hbar$, provided $\mathrm{Re}(\hbar) > 0$ (otherwise the large radius expansion embodied in the grand potential does not converge). We have explicitly verified this extended conjecture in many cases. Therefore, our analytic continuation of the trace to complex $\hbar$ matches the natural analytical continuation of the modified grand potential, in such a way that the conjecture of [18] remains true.

We now show that the minimal resurgent structure $\{\Phi_{0,n}(g_s), \Phi_{1,n}(g_s)\}$ can be recovered by performing a saddle point analysis of the state integral, but we hasten to comment that this is rather exceptional due to the particular nice form of the integrand in (3.48). In general, as we will see in Section 3.3.3, one cannot easily recover the minimal resurgent structure from the integral representation of the spectral traces.

In the limit $\hbar \to \infty$, $g_s = -1/\hbar \to 0$, the integrand has the semiclassical expansion

$$\exp \sum_{n=0}^{\infty} g_s^{2n-1} V_n(\tilde{x}, \mathrm{b}), \quad \tilde{x} = 2\pi \mathrm{b}^{-1} x, \tag{3.49}$$

where

$$V_0(\tilde{x}, \mathrm{b}) = -\frac{1}{2\pi^2 \mathrm{i}} \left( \pi \mathrm{i} \tilde{x} + 2\, \mathrm{Li}_2(-\mathrm{i} e^{\tilde{x}}) - 2\, \mathrm{Li}_2(\mathrm{i} e^{\tilde{x}}) \right). \tag{3.50}$$

The saddle point equation

$$\frac{\partial V_0(\tilde{x})}{\partial \tilde{x}} = -\frac{1}{2\pi^2 \mathrm{i}} \left( \pi \mathrm{i} - 2 \log \frac{1 + \mathrm{i} e^{\tilde{x}}}{1 - \mathrm{i} e^{\tilde{x}}} \right) = 0 \tag{3.51}$$

has two sets of solutions

$$\tilde{x} = \begin{cases} 2\pi \mathrm{i} \mathbb{Z}, \\ \pi \mathrm{i} + 2\pi \mathrm{i} \mathbb{Z}. \end{cases} \tag{3.52}$$

By expanding the integrand in the semiclassical limit around these saddle points, and performing Gaussian integration order by order in $g_s$, we obtain two infinite sets of asymptotic series. After proper normalisation they are precisely $\Phi_{0,n}(g_s)$ and $\Phi_{1,n}(g_s)$, as defined in (3.20), (3.22).[7] We want to point out that this in fact provides a very efficient way to compute the series $\Phi(g_s)$, and we are able to compute 400 terms of $\Phi(g_s)$.

Next, we demonstrate that the integral (3.48) can be evaluated explicitly by closing the contour from above and summing up residues, and show that the result factorises as a sum of products of holomorphic and anti-holomorphic blocks given by $q$ and $\tilde{q}$-series respectively, where

$$q = e^{2\pi \mathrm{i} \mathrm{b}^2}, \quad \tilde{q} = e^{-2\pi \mathrm{i} \mathrm{b}^{-2}}. \tag{3.53}$$

Note the definition of $q$ here is consistent with (3.36). Throughout the section we will assume that $\mathrm{Im}\, \mathrm{b}^2 > 0$ so that $|q|, |\tilde{q}| < 1$ and the $q, \tilde{q}$-series converge.

---

[7]If we plug $\tilde{x} = \pi \mathrm{i} + 2\pi \mathrm{i} \mathbb{Z}$ in $V_0(\tilde{x}, \mathrm{b})$ we seem to find the leading contribution $\exp(-\frac{1}{g_s}(\mathcal{V}_{\mathbb{F}_0} + \frac{\mathrm{i}}{2}))$ to $\Phi_1(g_s)$. However, as the singularity structure shown in Fig. 3 illustrates the correct leading contribution should be instead $\exp(-\frac{\mathcal{V}_{\mathbb{F}_0}}{g_s})$.

It turns out it is more convenient to first evaluate the integral (3.46) when $\xi \neq 0$, as the integrand of (3.46) has only simple poles, and then to evaluate the limit $\xi \to 0$. We assume that $\xi$ is small and since $\mathrm{Re}\,b > 0$, the poles in the upper half plane are

$$\pm \frac{b\xi}{2\pi} - \frac{ib}{4} + c_b + irb + isb^{-1}, \quad r,s = 0,1,2,\ldots, \tag{3.54}$$

where $c_b$ is defined in (A.4). The residues at these poles can be computed using (A.6), and by summing them up we find

$$\mathrm{Tr}\rho_{\mathbb{F}_0} = -\frac{1}{2}e^{-b^2\xi/2}q^{1/8}\frac{(q;q)_\infty^2(-\tilde{q}e^{2\xi};\tilde{q})_\infty(-\tilde{q}e^{-2\xi};\tilde{q})_\infty}{(q^{1/2};q)_\infty^2(\tilde{q}e^{2\xi};\tilde{q})_\infty(\tilde{q}e^{-2\xi};\tilde{q})_\infty}\coth(\xi)\left(\mathcal{I}(\xi,b) - \mathcal{I}(-\xi,b)\right) \tag{3.55}$$

where

$$\mathcal{I}(\xi,b) = e^{b^2\xi/2}\,_2\phi_1\left(\begin{matrix}q^{1/2}, q^{1/2}\\q\end{matrix}; q, q^{1/2}e^{2b^2\xi}\right)\,_2\phi_1\left(\begin{matrix}-1,-1\\\tilde{q}\end{matrix}; \tilde{q}, -\tilde{q}e^{-2\xi}\right) \tag{3.56}$$

and the $q$-hypergeometric function is defined in (A.22). Note that, as we explained in section 3.1, the holomorphic and the anti-holomorphic factors involve very different functions, in contrast to what happens in complex CS theory.

Before we turn off the mass parameter in the result, we notice that if we perform the shift $\xi \mapsto \xi + \pi i$, the variable $e^{2b^2\xi}$ in $\mathcal{I}(\xi,b)$ transforms by

$$e^{2b^2\xi} \mapsto qe^{2b^2\xi} \tag{3.57}$$

while the variable $e^{-2\xi}$ is invariant. This suggests to define the "massive" holomorphic blocks

$$A(x;q) = x^{1/4}\,_2\phi_1\left(\begin{matrix}q^{1/2}, q^{1/2}\\q\end{matrix}; q, q^{1/2}x\right), \tag{3.58a}$$

$$B(x;q) = x^{-1/4}\,_2\phi_1\left(\begin{matrix}q^{1/2}, q^{1/2}\\q\end{matrix}; q, q^{1/2}x^{-1}\right) \tag{3.58b}$$

$$= x^{-1/4}\frac{(q^{1/2};q)_\infty(qx^{-1};q)_\infty}{(q;q)_\infty(q^{1/2}x^{-1};q)_\infty}\,_2\phi_1\left(\begin{matrix}q^{1/2}, q^{1/2}x^{-1}\\qx^{-1}\end{matrix}; q, q^{1/2}\right), \tag{3.58c}$$

and "massive" anti-holomorphic blocks

$$\widetilde{A}(\tilde{x};\tilde{q}) = \,_2\phi_1\left(\begin{matrix}-1,-1\\\tilde{q}\end{matrix}; \tilde{q}, -\tilde{q}\tilde{x}^{-1}\right), \tag{3.59a}$$

$$\widetilde{B}(\tilde{x};\tilde{q}) = \widetilde{A}(\tilde{x}^{-1};\tilde{q}), \tag{3.59b}$$

where

$$x = e^{2b^2\xi}, \quad \tilde{x} = e^{2\xi}. \tag{3.60}$$

The second expression of $B(x;q)$ is obtained by applying Heine's transformation (A.23). Both holomorphic blocks satisfy the difference equation

$$(-1+q^{3/2}x)f(q^2x;q) + q^{1/4}(2-qx)f(qx;q) + q^{1/2}(-1+q^{1/2}x)f(x;q) = 0. \tag{3.61}$$

The Wronskian

$$W(x;q) = \det\begin{pmatrix}A(x;q) & B(x;q)\\A(qx;q) & B(qx;q)\end{pmatrix} \tag{3.62}$$

can be identified as

$$W(x;q) = q^{-1/4} \frac{(q^{1/2};q)^2_\infty (qx;q)_\infty (x^{-1};q)_\infty}{(q;q)^2_\infty (q^{1/2}x;q)_\infty (q^{-1/2}x^{-1};q)_\infty}. \tag{3.63}$$

We now take the massless limit $x \mapsto 1$. We find that

$$A(q^m e^u; q) = g_m(q) + \frac{1}{4} G_m(q)u + \mathcal{O}(u^2), \tag{3.64a}$$

$$B(q^m e^u; q) = h_m(q) - \frac{1}{4} H_m(q)u + \mathcal{O}(u^2), \tag{3.64b}$$

where

$$g_m(q) = q^{m/4} \sum_{n=0}^\infty \frac{(q^{1/2};q)^2_n}{(q;q)^2_n} q^{n/2+nm}, \quad m \geq 0, \tag{3.65a}$$

$$G_m(q) = q^{m/4} \sum_{n=0}^\infty \frac{(q^{1/2};q)^2_n}{(q;q)^2_n} q^{n/2+nm}(1+4n), \quad m \geq 0, \tag{3.65b}$$

as well as

$$h_m(q) = q^{m/4} \sum_{n=0}^\infty \frac{(q^{1/2};q)_{n+m}(q^{1/2};q)_n}{(q;q)_{n+m}(q;q)_n} q^{n/2}, \qquad m \geq 0, \tag{3.66a}$$

$$H_m(q) = -4q^{-m/4} \sum_{n=0}^{m-1} \frac{(q^{1/2};q)_n (q^{-1};q^{-1})_{m-n-1}}{(q;q)_n (q^{-1/2};q^{-1})_{m-n}} q^{n/2} \qquad m \geq 0$$
$$+ q^{m/4} \sum_{n=0}^\infty \frac{(q^{1/2};q)_{n+m}(q^{1/2};q)_n}{(q;q)_{n+m}(q;q)_n} q^{n/2} \left(1 - 4\sum_{j=1}^\infty \left(\frac{q^{j+n}}{1-q^{j+n}} - \frac{q^{j-1/2+n}}{1-q^{j-1/2+n}}\right)\right). \tag{3.66b}$$

We comment that when expanding $B(q^m e^u; q)$ we need to use the second expression of $B(x;q)$ in (3.58c), since in the first expression of $B(q^m e^u; q)$ with $m \geq 1$, the power of $q$ is not bounded from below. In addition, we find by explicit $q$-expansion that

$$h_m(q) = g_m(q), \quad G_0(q) = H_0(q). \tag{3.67}$$

The series $g_m(q)$, $G_m(q)$, $h_m(q)$ and $H_m(q)$ in (3.64) with $m > 0$ are the analogues of the "descendants" introduced in [11, 12]. As we will see, although they do not appear in the expression for the spectral trace, they are necessary to reconstruct the Stokes data. For the anti-holomorphic block we find the expansion

$$\widetilde{A}(e^{\tilde{u}}; \tilde{q}) = 2\tilde{g}_0(\tilde{q}) + 4\tilde{u}\, \tilde{G}_0(\tilde{q}) + \mathcal{O}(\tilde{u}^2), \tag{3.68}$$

where

$$\tilde{g}_0(\tilde{q}) = \frac{1}{2} \sum_{n=0}^\infty \frac{(-1;\tilde{q})^2_n}{(\tilde{q};\tilde{q})^2_n}(-\tilde{q})^n, \tag{3.69a}$$

$$\tilde{G}_0(\tilde{q}) = -\frac{1}{4} \sum_{n=0}^\infty \frac{(-1;\tilde{q})^2_n}{(\tilde{q};\tilde{q})^2_n}(-\tilde{q})^n n. \tag{3.69b}$$

As an application of these results, we can take the massless limit of the factorisation formula (3.55) and find

$$\mathrm{Tr}\rho_{\mathbb{F}_0} = -\frac{i}{2}\left(G_0(q)\tilde{g}_0(\tilde{q}) + 8b^{-2}g_0(q)\tilde{G}_0(\tilde{q})\right). \tag{3.70}$$

Note that this is very similar to the factorization in holomorphic blocks of the state integral invariant of the figure-eight knot in [75]. However, as noted above, the factorization is not symmetric, and the holomorphic and anti-holomorphic blocks are given by different series. Another important result is that in the massless limit the Wronskian identity (3.63) implies that

$$g_0(G_1 + H_1) - 2g_1 G_0 = \frac{4q^{1/4}}{1 - q^{1/2}}. \tag{3.71}$$

### 3.2.3 Radial asymptotic analysis

One of the important lessons we learned from [11, 12] is that it is often helpful to study the asymptotic behavior of the anti-holomorphic blocks[8] in the limit $g_s \propto -b^{-2} \to 0$ along a ray $e^{i\theta}\mathbb{R}_+$, which depends crucially on the angle $\theta$. Oftentimes, it is possible to promote these radial asymptotic behavior of anti-holomorphic blocks to exact representations of them in terms of the Borel resummation of the asymptotic series $\Phi_\sigma(g_s)$, together with non-perturbative corrections parametrised by $q$-series. This turns out to be an extremely powerful method to compute the Stokes automorphism of the asymptotic series $\Phi_\sigma(g_s)$.

In the case of the anti-holomorphic blocks $\tilde{g}(\tilde{q}), \tilde{G}(\tilde{q})$ uncovered from evaluating the first trace of local $\mathbb{F}_0$, we find that they can be expressed in terms of Borel resummations of $\Phi(g_s)$ in the following way

$$\begin{pmatrix} \frac{1}{\sqrt{\pi g_s}} \tilde{g}(\tilde{q}) \\ -8\sqrt{\pi g_s} \tilde{G}(\tilde{q}) \end{pmatrix} = 2^{-3/2} M_R(q) s_R(\Phi)(g_s), \tag{3.72}$$

where $M_R(q)$ is a $2 \times 2$ matrix of $q$-series that encodes non-perturbative corrections. We parametrise it by

$$M_R(q) = \begin{pmatrix} g_+^R(q) & -g_-^R(q) \\ G_+^R(q) & G_-^R(q) \end{pmatrix}. \tag{3.73}$$

Note that it depends on the sector $R$. We will focus on the two sectors $I, II$, as shown in Fig. 4.

When $g_s$ is in the sector $I$, we find that

$$g_+^I(q) = 1 + q^{1/2} - q + 2q^{3/2} - 2q^2 + \dots, \tag{3.74a}$$

$$G_+^I(q) = 1 + 5q^{1/2} - q + 10q^{3/2} - 2q^2 + \dots, \tag{3.74b}$$

$$g_-^I(q) = 1 + 3q^{1/2} - 2q + 3q^{3/2} + \dots, \tag{3.74c}$$

$$G_-^I(q) = 1 - 9q^{1/2} - 2q - 9q^{3/2} + \dots. \tag{3.74d}$$

it is easy to identify that

$$g_+^I(q) = g_0(q), \quad G_+^I(q) = G_0(q), \tag{3.75}$$

and we will denote them by $g(q), G(q)$. In addition, these four $q$-series satisfy the "Wronskian"-like identity

$$g_+^I(q) G_-^I(q) + G_+^I(q) g_-^I(q) = 2. \tag{3.76}$$

With the help of (3.71), we are able to identify

$$g_-^I(q) = -\frac{1 - q^{1/2}}{q^{1/4}} g_1(q) + 2g_0(q), \tag{3.77a}$$

$$G_-^I(q) = \frac{1 - q^{1/2}}{2q^{1/4}} (G_1(q) + H_1(q)) - 2G_0(q), \tag{3.77b}$$

---

[8]In [11,12] one studies the asymptotics of holomorphic blocks in the opposite limit $b^2 \to 0$, which corresponds to the semiclassical regime.

which we will denote by $g'(q), G'(q)$. Similarly in sector $II$, we find in fact

$$\begin{aligned} g_-^{II}(q) &= g(q), \quad G_-^{II}(q) = G(q), \\ g_+^{II}(q) &= g'(q), \quad G_+^{II}(q) = G'(q). \end{aligned} \tag{3.78}$$

Once $M_{I,II}(q)$ are known, the Stokes automorphism $\mathfrak{S}_{I\mapsto II}(q)$ can be computed by

$$\mathfrak{S}_{I\mapsto II}(q) = M_{II}(q)^{-1}M_I(q), \tag{3.79}$$

which yields (3.34).

This calculation illustrates that the additional analytic structures provided by the TS/ST correspondence makes it possible to calculate the Stokes constants associated to the series $\Phi_{0,1}(g_s)$ and to express it in closed form, in terms of the $q$-series appearing in the factorization of the spectral trace (3.71) and their descendants (3.64).

We close this subsection by commenting that the radial asymptotic formula (3.72) together with the factorisation formula (3.70) allows us to express the trace in terms of Borel sum of asymptotic series:

- When $g_s$ is in sector $I$

$$\mathrm{Tr}\rho_{\mathbb{F}_0} = -\frac{\mathrm{i}\sqrt{\pi g_s}}{2^{3/2}}\left( g(q)G(q)s_I(\Phi_0)(g_s) + \frac{1}{2}(g(q)G'(q) - G(q)g'(q))s_I(\Phi_1)(g_s)\right). \tag{3.80}$$

  Note that here $\Phi_0(g_s)$ is the dominant series.

- When $g_s$ is in sector $II$

$$\mathrm{Tr}\rho_{\mathbb{F}_0} = -\frac{\mathrm{i}\sqrt{\pi g_s}}{2^{3/2}}s_{II}(\Phi_0)(g_s). \tag{3.81}$$

In both sectors in the leading order we have

$$Z_1(\hbar) = \mathrm{Tr}\rho_{\mathbb{F}_0} \sim -\frac{\mathrm{i}\sqrt{\pi g_s}}{2^{3/2}}\Phi_0(g_s), \quad |g_s| \ll 1, \tag{3.82}$$

which is consistent with the prediction (3.5) from the TS/ST correspondence.

### 3.2.4 Relation to $q$-Painlevé

In [79], it has been conjectured that the spectral determinant of local $\mathbb{F}_0$ satisfies a $q$-deformed Painlevé equation, which is a difference equation involving the mass parameter $\xi$. This leads to $q$-difference equations for the spectral traces. Since we have an explicit expression for the first trace in terms of $q$-hypergeometric functions given by (3.55), it is natural that the $q$-Painlevé equation maps to the $q$-difference equation satisfied by the $q$-hypergeometric function. We will now explicitly show that this is the case.

In the notation used above, the $q$-difference equation found in [79] reads

$$2\tanh(\xi)Z(\xi) + Z(\xi + \pi\mathrm{i}\mathrm{b}^{-2}) + Z(\xi - \pi\mathrm{i}\mathrm{b}^{-2}) = 0. \tag{3.83}$$

First of all, we note that this relation does not affect the "holomorphic" part of the trace, since the above shifts leave invariant $\mathrm{e}^{2\mathrm{b}^2\xi}$. In terms of the dual mass parameter

$$\xi_D = -\mathrm{b}^2\xi \tag{3.84}$$

and the exponentiated variables

$$z = \mathrm{e}^{-2\xi}, \qquad z_D = \mathrm{e}^{-2\xi_D}, \tag{3.85}$$

we can write down the factorization

$$\frac{(q;q)_\infty^2(-\tilde{q}e^{2\xi};\tilde{q})_\infty(-\tilde{q}e^{-2\xi};\tilde{q})_\infty}{(q^{1/2};q)_\infty^2(\tilde{q}e^{2\xi};\tilde{q})_\infty(\tilde{q}e^{-2\xi};\tilde{q})_\infty}\coth(\xi)\mathcal{I}(\xi,b) = \mathcal{H}(q,z_D)\widetilde{\mathcal{H}}(\tilde{q},z),\tag{3.86}$$

where

$$\begin{aligned}
\mathcal{H}(q,z_D) &= e^{b^2\xi/2}\frac{(qe^{2b^2\xi};q)_\infty(q;q)_\infty}{(q^{1/2};q)_\infty(q^{1/2}e^{2b^2\xi};q)_\infty}{}_2\phi_1\left(\begin{matrix}q^{1/2},q^{1/2}e^{2b^2\xi}\\qe^{2b^2\xi}\end{matrix};q,q^{1/2}\right),\\
\widetilde{\mathcal{H}}(\tilde{q},z) &= \coth(\xi)\frac{(-\tilde{q}e^{2\xi};\tilde{q})_\infty(-\tilde{q}e^{-2\xi};\tilde{q})_\infty}{(\tilde{q}e^{2\xi};\tilde{q})_\infty(\tilde{q}e^{-2\xi};\tilde{q})_\infty}{}_2\phi_1\left(\begin{matrix}-1,-1\\\tilde{q}\end{matrix};\tilde{q},-\tilde{q}e^{-2\xi}\right).
\end{aligned}\tag{3.87}$$

We can write

$$Z(\xi) = -\frac{z_D^{-1/4}q^{1/8}}{2}\left(\mathcal{H}(q,z_D)\widetilde{\mathcal{H}}(\tilde{q},z) - \mathcal{H}(q,z_D^{-1})\widetilde{\mathcal{H}}(\tilde{q},z^{-1})\right).\tag{3.88}$$

We note that

$$Z(\xi\pm\pi i b^{-2}) = -\frac{z_D^{-1/4}q^{1/8}}{2}\left(\mathcal{H}(q,z_D)\widetilde{\mathcal{H}}(\tilde{q},z\tilde{q}^{\mp1}) + \mathcal{H}(q,z_D^{-1})\widetilde{\mathcal{H}}(\tilde{q},z^{-1}\tilde{q}^{\pm1})\right).\tag{3.89}$$

The difference equation (3.83) implies then two equations,

$$\begin{aligned}
-2\widetilde{\mathcal{H}}(\tilde{q},z) &= \frac{\tilde{q}+z}{\tilde{q}-z}\widetilde{\mathcal{H}}(\tilde{q},z/q) + \frac{1+\tilde{q}z}{1-\tilde{q}z}\widetilde{\mathcal{H}}(\tilde{q},zq),\\
2\widetilde{\mathcal{H}}(\tilde{q},1/z) &= \frac{\tilde{q}+z}{\tilde{q}-z}\widetilde{\mathcal{H}}(\tilde{q},q/z) + \frac{1+\tilde{q}z}{1-\tilde{q}z}\widetilde{\mathcal{H}}(\tilde{q},1/zq),
\end{aligned}\tag{3.90}$$

which are simply exchanged by $z\leftrightarrow 1/z$. Let us then focus on the first one. By plugging in the explicit expression for $\mathcal{H}(\tilde{q},z)$, and changing the sign $z\to -z$, we find the simple difference equation

$$2\phi(\tilde{q}z) = \frac{1-z}{1+z}\left(\phi(z)+\phi(z\tilde{q}^2)\right),\tag{3.91}$$

where

$$\phi(z) = {}_2\phi_1\left(\begin{matrix}-1,-1\\\tilde{q}\end{matrix};\tilde{q},z\right).\tag{3.92}$$

On the other hand, a generic $q$-hypergeometric function

$$\Phi = {}_2\phi_1\left(\begin{matrix}a,b\\c\end{matrix};q,z\right)\tag{3.93}$$

satisfies the following $q$-difference equation[9]

$$z(c-abqz)\mathcal{D}_q^2\Phi + \left(\frac{1-c}{1-q} + \frac{(1-a)(1-b)-(1-abq)}{1-q}z\right)\mathcal{D}_q\Phi - \frac{(1-a)(1-b)}{(1-q)^2}\Phi = 0,\tag{3.94}$$

where

$$\mathcal{D}f(z) = \frac{f(z)-f(zq)}{(1-q)z}.\tag{3.95}$$

It is easy to see that, when $a=b=-1$, $c=q$, the equation (3.94) becomes (3.91), after setting $q\to\tilde{q}$.

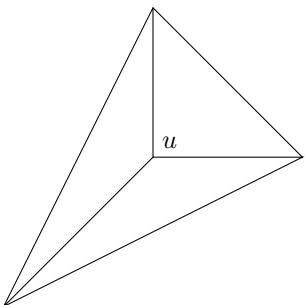

Figure 6: The toric diagram of the local $\mathbb{P}^2$ geometry.

## 3.3 Local $\mathbb{P}^2$

The next example we consider is the canonical bundle over the surface $\mathbb{P}^2$, which is also called the local $\mathbb{P}^2$ geometry,

$$X = \mathcal{O}(-3) \to \mathbb{P}^2. \tag{3.96}$$

This geometry is also toric and it is described by the toric diagram in Fig. 6. There is one true Kähler modulus corresponding to the internal vertex of the toric diagram, and no mass parameter. For the topological string theory on local $\mathbb{P}^2$, the full moduli space is identified with the family of mirror curves described by the equation

$$e^x + e^y + e^{-x-y} + \tilde{u} = 0, \quad x, y \in \mathbb{C}, \tag{3.97}$$

and it is parametrised by $u = 1/\tilde{u}^3$. The maximal conifold point on the moduli space is at the discriminant locus of (3.97)

$$u = -\frac{1}{27}. \tag{3.98}$$

The free energies of the topological string at the maximal conifold point were computed in [42, 86] by using the holomorphic anomaly equations [84, 85]. The series for $\mathcal{F}_0(\lambda)$ was written down in (2.6). For $g = 1, 2$, one finds

$$\mathcal{F}_1(\lambda) = -\frac{1}{12}\log(\lambda) + \frac{5\pi^2\lambda}{18\sqrt{3}} - \frac{\pi^4\lambda^2}{486} - \frac{40\pi^6\lambda^3}{2187\sqrt{3}} + \frac{283\pi^8\lambda^4}{32805} + \dots, \tag{3.99a}$$

$$\mathcal{F}_2(\lambda) = -\frac{1}{240\lambda^2} + \frac{4\pi^6\lambda}{405\sqrt{3}} - \frac{3187\pi^8\lambda^2}{492075} + \dots. \tag{3.99b}$$

We can then construct the formal power series $\Phi_{0;N}(g_s)$ by (2.11).

In this section we will study the resurgent structure of the power series for both $N = 1$ and $N = 2$ of the local $\mathbb{P}^2$ geometry.

### 3.3.1 The first resurgent structure

From the free energies $\mathcal{F}_g(\lambda)$ we assemble the series for $N = 1$

$$\Phi_{0;1}(g_s) = \exp\left(\frac{\mathcal{V}_{\mathbb{P}^2}}{g_s}\right) \varphi_{0;1}(g_s), \tag{3.100}$$

with

$$\mathcal{V}_{\mathbb{P}^2} = \frac{3V}{4\pi^2} \tag{3.101}$$

---

[9]See for instance http://dlmf.nist.gov/17.6.E27.

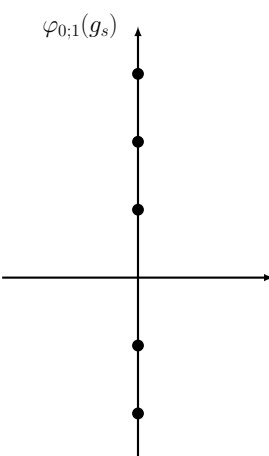

Figure 7: The singularities in the Borel plane for the series $\varphi_{0;1}(g_s)$.

and

$$\varphi_{0;1}(g_s) = 1 - \frac{\pi^2 g_s}{6\sqrt{3}} + \frac{\pi^4 g_s^2}{216} + \frac{59\pi^6 g_s^3}{19440\sqrt{3}} - \frac{251\pi^8 g_s^4}{1399680} - \frac{23687\pi^{10}g_s^5}{58786560\sqrt{3}} + \frac{785699\pi^{12}g_s^6}{31744742400} + \cdots.$$
(3.102)

The resurgent structure of $\Phi_{0;1}(g_s)$ is very simple. The Borel transform $\widehat{\Phi}_{0;1}(\zeta)$ has infinitely many singularities along the imaginary axis located at

$$\mathrm{i}n, \quad n \in \mathbb{Z}_{\neq 0},$$
(3.103)

as illustrated in Fig. 7. They correspond to the asymptotic series in the same family as $\Phi_{0;1}(g_s)$

$$\Phi_{0,n;1}(g_s) = \Phi_{0;1}(g_s)\mathrm{e}^{-n\frac{\mathrm{i}}{g_s}}, \quad n \in \mathbb{Z}.$$
(3.104)

We conclude that the minimal resurgent structure associated to $\Phi_{0;1}(g_s)$ is the set $\{\Phi_{0,n;1}(g_s)\}_{n\in\mathbb{Z}}$.

Following the structure of Borel singularities, there is a single generating series of Stokes constants

$$\mathsf{S}_{00;1}(q) = \sum_{n\in\mathbb{Z}_{\neq 0}} \mathsf{S}_{00,n;1}q^{n/3}.$$
(3.105)

Note that we have scaled the power of $q$ by a factor of $1/3$, as compared to (2.34), similarly to what we did in the example of $\mathbb{F}_0$. The meaning of this will be clear in the next subsection. The generating function can be decomposed as

$$\mathsf{S}_{00;1}(q) = \mathsf{S}_1^+(q) + \mathsf{S}_1^-(q),$$
(3.106)

where $\mathsf{S}_1^{\pm}(q)$ are $q$ and $q^{-1}$-series respectively, encoding the Stokes constants in the upper and lower half planes. The identity

$$\Phi_{0;1}(g_s)^* = \Phi_{0;1}(g_s^*),$$
(3.107)

implies that

$$\mathsf{S}_1^-(q) = (1 + \mathsf{S}_1^+(q^{-1}))^{-1} - 1,$$
(3.108)

and therefore we only have to compute $\mathsf{S}_1^+(q)$.

To compute the $q$-series $\mathsf{S}_1^+(q)$, we invoke the TS/ST correspondence [18,19]. The integral kernel for the local $\mathbb{P}^2$ geometry is [59]

$$\rho_{\mathbb{P}^2}(x_1, x_2) = \frac{\mathrm{e}^{2\pi a(x_1+x_2)}}{2\mathsf{b}\cosh\pi\frac{x_1-x_2+\mathrm{i}h}{\mathsf{b}}} \frac{\Phi_{\mathsf{b}}(x_2 + 2\mathrm{i}a)}{\Phi_{\mathsf{b}}(x_1 - 2\mathrm{i}a)},$$
(3.109)

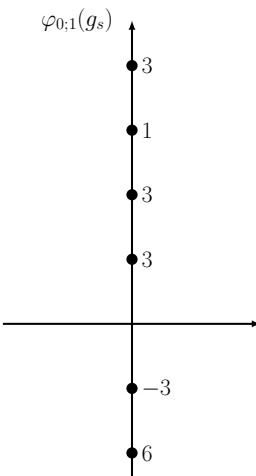

Figure 8: Stokes constants for the asymptotic series from the first trace of local $\mathbb{P}^2$.

where

$$a = h = \frac{b}{6}, \tag{3.110}$$

and the Planck constant $\hbar$ is related to b by

$$\hbar = \frac{2\pi b^2}{3}. \tag{3.111}$$

As in the case of local $\mathbb{F}_0$, we can use the explicit expression (3.109) to extend the integral kernel to complex values of $\hbar$ in $\mathbb{C}'$, corresponding to $\mathrm{Re}\, b > 0$. The first "bosonic" trace has the integral form [59]

$$\mathrm{Tr}\rho_{\mathbb{P}^2} = \frac{1}{2b\cos(\pi h/b)} \int_{\mathbb{R}} e^{4\pi xa} \frac{\Phi_b(x + 2ia)}{\Phi_b(x - 2ia)} dx, \tag{3.112}$$

which is also identified with the fermionic trace $Z_1(\hbar)$. This integral can be quickly evaluated by either using the integral Ramanujan formula or by completing the integration contour from above, and it yields [59]

$$\mathrm{Tr}\rho_{\mathbb{P}^2} = \frac{1}{\sqrt{3}b} e^{\frac{\pi i}{12}(b^2 + b^{-2}) + \frac{\pi i}{4} - \frac{\pi i}{9}b^2} \frac{\Phi_b(c_b - \frac{ib}{3})^2}{\Phi_b(c_b - \frac{2ib}{3})}. \tag{3.113}$$

Note that in the limit $\hbar \to \infty$, or $g_s = -1/\hbar \to 0$, (3.113) is asymptotically

$$\mathrm{Tr}\rho_{\mathbb{P}^2} \sim -i3^{-3/4}(2\pi g_s/3)^{1/2}\Phi_{0;1}(g_s), \tag{3.114}$$

and therefore the relation (3.5) is upheld.

More is actually true. By high precision numerical calculation, we verify that, when $g_s$ is in the second quadrant, the Borel resummation of $\Phi_{0;1}(g_s)$ is *identical* to the first spectral trace, up to a simple prefactor

$$\mathrm{Tr}\rho_{\mathbb{P}^2} = -i3^{-3/4}(2\pi g_s/3)^{1/2}s_{II}(\Phi_{0;1})(g_s). \tag{3.115}$$

On the other hand, when $g_s$ is in the first quadrant

$$\mathrm{Tr}\rho_{\mathbb{P}^2} = -i3^{-3/4}(2\pi g_s/3)^{1/2}s_I(\Phi_{0;1})(g_s)K_1(q), \tag{3.116}$$

where

$$K_1(q) = \frac{(q^{2/3}; q)_\infty^3}{(q^{1/3}; q)_\infty^3},$$ (3.117)

with

$$q = e^{2\pi i b^2} = e^{-3i/g_s}.$$ (3.118)

Note that, here, $q$ is defined in such a way that $q^{1/3}$ accounts for the distance between neighboring singularities in the tower (3.103). Therefore,

$$S_1^+(q) = K_1(q) - 1 = 3q^{1/3} + 3q^{2/3} + q + 3q^{4/3} + 6q^{5/3} - 3q^{7/3} + 9q^{8/3} + 9q^3 + \dots. \quad (3.119)$$

Using (3.108) we find the Stokes constants in the lower half plane

$$S_1^-(q) = -3q^{-1/3} + 6q^{-2/3} - 10q^{-1} + 12q^{-4/3} - 9q^{-5/3} + q^{-2} + \dots. \quad (3.120)$$

These Stokes constants are illustrated in Fig. 8.

### 3.3.2 The second resurgent structure

We can use the conifold free energies to construct the formal power series $\Phi_{0;2}(g_s)$ at $N = 2$. Instead we will consider the following normalised series[10]

$$\begin{aligned}
\Phi'_{0;2}(g_s) &= 1 + \frac{8\pi^2 g_s}{9\sqrt{3}} \frac{\Phi_{0;2}(g_s)}{\Phi_{0;1}(g_s)^2} = \sum_{n \geq 0} a_n g_s^n \\
&= 1 + \frac{8\pi^2 g_s}{9\sqrt{3}} + \frac{16\pi^4 g_s^2}{81} + \frac{64\pi^6 g_s^3}{729\sqrt{3}} - \frac{64\pi^8 g_s^4}{19683} + \dots
\end{aligned}$$ (3.121)

Up to normalisation, this is the power series associated to the normalised second trace

$$\mathrm{Tr}'\rho_{\mathbb{P}^2}^2 := \frac{\mathrm{Tr}\rho_{\mathbb{P}^2}^2}{(\mathrm{Tr}\rho_{\mathbb{P}^2})^2} = 1 - \frac{2Z_2(\hbar)}{Z_1(\hbar)^2}.$$ (3.122)

As we will see, the advantage of working with this series is that its resurgent structure is simpler. In the remainder of this section and the following sections we will drop the prime and always refer to the normalised series (3.121) when we use the symbol $\Phi_{0;2}(g_s)$. We will refer to $\mathfrak{B}_{\Phi_{0;2}}$ as the second resurgent structure of local $\mathbb{P}^2$.

The resurgent structure associated to $\Phi_{0;2}(g_s)$ is quite interesting. The Borel transform $\widehat{\Phi}_{0;2}(\zeta)$ has one family of infinitely many singularities in the first and the fourth quadrants as illustrated in Fig. 9, and they are located at

$$3\mathcal{V}_{\mathbb{P}^2} + in, \quad n \in \frac{1}{2} + \mathbb{Z}.$$ (3.123)

This indicates that there exists a second family of asymptotic series, which we denote by

$$\Phi_{1,n;2}(g_s) = \Phi_{1;2}(g_s) e^{-n \frac{i}{g_s}}, \quad n \in \frac{1}{2} + \mathbb{Z}.$$ (3.124)

Note that here we have generalized the convention (2.32) in Section 2.3 in order to emphasize the reflection symmetry of the singularities in the Borel plane. In (3.124),

$$\Phi_{1;2}(g_s) = e^{-\frac{3\mathcal{V}_{\mathbb{P}^2}}{g_s}} \varphi_{1;2}(g_s),$$ (3.125)

---

[10]The fact that the normalised series has no exponential factor indicates that the leading contribution to $\Phi_{0;2}(g_s) \sim Z_2(\hbar)$ is indeed $\exp(2\mathcal{V}_{\mathbb{P}^2}/g_s)$ as predicted by (3.8).

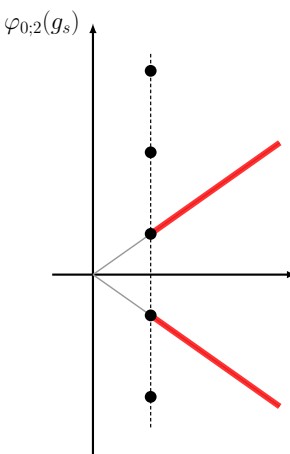

Figure 9: The singularities in the Borel plane for the series $\varphi_{0;2}(g_s)$

and we assume $\varphi_{1;2}(g_s)$ has the general form

$$\varphi_{1;2}(g_s) = (-g_s)^{-\nu} \sum_{n=0}^{\infty} a_{n,1} g_s^n. \tag{3.126}$$

To uncover the nature of the series $\varphi_{1;2}(g_s)$, we focus on the two nearest singularities

$$\zeta_1 := 3\mathcal{V}_{\mathbb{P}^2} + \frac{i}{2}, \quad \zeta_1^* = 3\mathcal{V}_{\mathbb{P}^2} - \frac{i}{2}, \tag{3.127}$$

which are conjugate to each other. It follows from standard resurgent analysis that the perturbative series $\varphi_{1;2}(g_s)$ at these two singular points control the large order behavior of the coefficients $a_n$ of the series $\Phi_{0,2}(g_s)$, albeit in a more complicated form

$$a_n \sim \frac{S_{01,0;2}}{\pi} \sum_{k=0}^{\infty} |\zeta_1|^{k-n-\nu} \Gamma(n+\nu-k)|a_{k,1}| \sin(\theta_k - (n+\nu-k)\theta). \tag{3.128}$$

Here,

$$\theta = \arg \zeta_1, \quad \theta_k = \arg a_{k,1}. \tag{3.129}$$

Therefore, it should be possible to extract the coefficients $a_{k,1}$ from the large order behavior of the series $a_n$. Since the asymptotics is oscillatory, one can not apply the standard Richardson transform which is often used in resurgent analysis. It turns out that, for this type of oscillatory behavior, there is a numerical algorithm due to Hunter and Guerrieri [87] which makes it possible to extract the coefficients $a_{k,1}$ (see Appendix C for an explanation of this algorithm). By using this algorithm, we find

$$S_{01,0;2}\varphi_{1;2}(g_s) = -2^{1/2}3^{9/4}\pi^{-1/2}g_s^{-1/2}\exp\left(\frac{\pi^2 g_s}{2\sqrt{3}} - \frac{4\pi^6 g_s^3}{405\sqrt{3}} + \dots\right), \tag{3.130}$$

which turns out to be proportional to $\Phi_{0;1}(g_s)^{-3}$. We choose the normalisation $S_{01,0;2} = 1$ so that the second asymptotic series associated to the normalised second trace reads

$$\Phi_{1;2}(g_s) = -2^{1/2}3^{9/4}\pi^{-1/2}g_s^{-1/2}\Phi_{0;1}(g_s)^{-3}. \tag{3.131}$$

Since by Stokes automorphism $\Phi_{0;1}(g_s)$, and hence $\Phi_{1;2}(g_s)$, transforms back to itself, the two families of power series $\{\Phi_{0,n;2}(g_s), \Phi_{1,n;2}(g_s)\}$ form a minimal resurgent structure. The

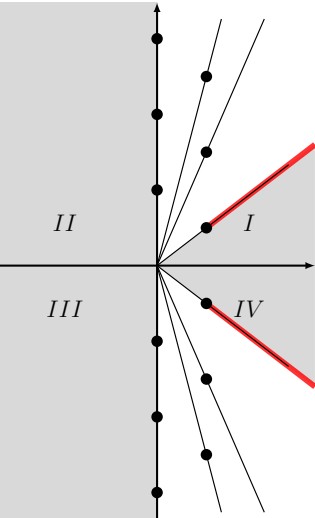

Figure 10: Stokes rays and sectors in the $g_s$-plane for the series $\varphi_{0;2}(g_s), \varphi_{1;2}(g_s)$ in the second resurgent structure of local $\mathbb{P}^2$.

complete set of Borel singularities of this minimal resurgent structure is illustrated in Fig. 10. Stokes rays passing through these singular points divide the complex plane into infinitely many sectors in the first and the fourth quadrants. As usual we denote the sectors bordering the positive real axis in the first and the fourth quadrants as $I, IV$ and sometimes refer to the second and the third quadrants as sectors $II, III$.

Following the pattern of Stokes rays we can collect all the Stokes constants in the $2 \times 2$ Stokes matrix

$$\mathsf{S}_2(q) = \begin{pmatrix} 0 & \mathsf{S}_{01;2}(q) \\ 0 & \mathsf{S}_{11;2}(q) \end{pmatrix}, \tag{3.132}$$

and then decompose it as

$$\mathsf{S}_2(q) = \mathsf{S}_2^+(q) + \mathsf{S}_2^-(q), \tag{3.133}$$

where $\mathsf{S}_2^+(q)$ and $\mathsf{S}_2^-(q)$ are respectively $q$- and $q^{-1}$-series encoding Stokes constants in the upper and lower half planes. Here we choose for $q$ the same convention as in the previous subsection

$$q = e^{2\pi i b^2} = e^{3i\hbar} = e^{-\frac{3i}{g_s}}, \tag{3.134}$$

so that the entries of $\mathsf{S}_2(q)$ are

$$\mathsf{S}_{01;2}(q) = q^{\frac{1}{6}} \sum_{n=0}^{\infty} \mathsf{S}_{01,n;2} q^{\frac{n}{3}}, \quad \mathsf{S}_{11;2}(q) = \sum_{n=1}^{\infty} \mathsf{S}_{11,n;2} q^{\frac{n}{3}}. \tag{3.135}$$

Since under complex conjugation

$$\Phi_{-;2}(g_s)^* = \Phi_{-;2}(g_s^*), \tag{3.136}$$

it follows that

$$\mathsf{S}_2^-(q) = (\mathbf{1} + \mathsf{S}_2^+(q^{-1}))^{-1} - \mathbf{1}, \tag{3.137}$$

and we only have to compute $\mathsf{S}_2^+(q)$. The component $\mathsf{S}_{11;2}^+(q)$ can be immediately identified

$$\mathsf{S}_{11;2}^+(q) = (1 + \mathsf{S}_1^+(q))^{-3} - 1 = -9q^{1/3} + 45q^{2/3} - 165q + 486q^{4/3} - 1197q^{5/3} + 2517q^2 + \dots. \tag{3.138}$$

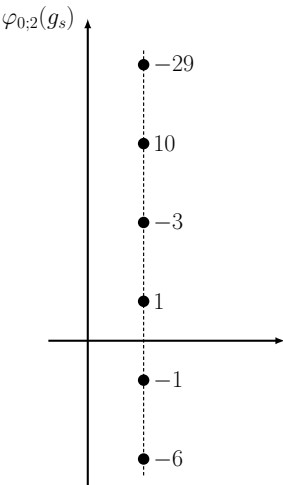

Figure 11: Off-diagonal Stokes constants for the second resurgent structure of local $\mathbb{P}^2$.

The first term of $S^+_{01;2}(q)$ is $q^{1/6}$ by our normalisation. More terms of $S^+_{01;2}(q)$ can again be computed by using the TS/ST correspondence and radial asymptotic analysis, to which we will momentarily turn. We end this subsection by recording the results here

$$S^+_{01;2}(q) = q^{1/6}(1 - 3q^{1/3} + 10q^{2/3} - 29q + 72q^{4/3} - 155q^{5/3} + 291q^2 - 474q^{7/3}$$
$$+ 660q^{8/3} - 760q^3 + 663q^{10/3} - 309q^{11/3} - 193q^4 + \ldots). \tag{3.139}$$

Using (3.137) we can write down the following Stokes constants in the lower half plane as well:

$$S^-_{11;2}(q) = (1 + S^+_1(q^{-1}))^3 - 1 = 9q^{-1/3} + 36q^{-2/3} + 84q^{-1} + 135q^{-4/3}$$
$$+ 198q^{-5/3} + 327q^{-2} + \ldots,$$
$$S^-_{01;2}(q) = -S^+_{01;2}(q^{-1})/(1 + S^+_{11;2}(q^{-1}))$$
$$= -q^{-1/6}(1 + 6q^{-1/3} + 19q^{-2/3} + 37q^{-1} + 54q^{-4/3} + 82q^{-5/3} + 135q^{-2} + 174q^{-7/3}$$
$$+ 171q^{-8/3} + 234q^{-3} + 399q^{-10/3} + 406q^{-11/3} + 273q^{-4} + \ldots). \tag{3.140}$$

The diagonal Stokes constants in the series $S_{11;2}(q)$ are responsible for the automorphism of $\Phi_{1;2}(g_s)$ back to itself and they are similar to Stokes constants in $S_{00;1}(q)$ which are illustrated in Fig. 8. The off-diagonal Stokes constants in the series $S_{01;2}(q)$ are illustrated in Fig. 11.

### 3.3.3 Second spectral trace and its factorisation

The TS/ST correspondence allows us to write down an integral representation of the second trace. In principle the second bosonic trace $\mathrm{Tr}\rho^2_{\mathbb{P}^2}$ involves a double integral of a product of the integral kernel (3.109), but it was shown in [59] that it can be converted into a one-dimensional integral. Using the result of the first trace (3.113), this translates into the following integral representation of the normalised second trace

$$\mathrm{Tr}'\rho^2_{\mathbb{P}^2} = \sqrt{3}\mathsf{b}\frac{\Phi_\mathsf{b}(-\frac{i\mathsf{b}}{6} - \frac{i\mathsf{b}^{-1}}{2})^2}{\Phi_\mathsf{b}(\frac{i\mathsf{b}}{6} + \frac{i\mathsf{b}^{-1}}{2})^2} \int_\mathbb{R} \frac{\sinh(2\pi hx)}{\sinh(\pi\mathsf{b}x)} \frac{\Phi_\mathsf{b}(x - \frac{i\mathsf{b}}{3} + c_\mathsf{b})^2}{\Phi_\mathsf{b}(x + \frac{i\mathsf{b}}{3} - c_\mathsf{b})^2} e^{2\pi x(\frac{1}{3}\mathsf{b}+\mathsf{b}^{-1})}dx. \tag{3.141}$$

In order to perform the large b expansion of the integral (3.141), one can first do the scaling $x \to \mathsf{b}^{-1}x$ and then use the formula of semi-classical expansion of the quantum dilogarithm

(A.16) for large b (or equivalently (A.15) for small $b_D = 1/b$). After evaluating convergent integrals

$$\int_{\mathbb{R}} \frac{\sinh(\frac{1}{3}\pi x)}{\sinh(\pi x)} x^{2n} \mathrm{d}x = \frac{2(-1)^{n-1}}{2n+1} B_{2n+1}(\tfrac{1}{3}) 3^{2n+1/2}, \quad n = 0, 1, 2, \dots, \quad (3.142)$$

one finds that

$$\mathrm{Tr}' \rho_{\mathbb{P}^2}^2 \sim \Phi_{0;2}(g_s) = 1 + \frac{8\pi^2}{9\sqrt{3}} g_s + \frac{16\pi^4}{81} g_s^2 + \dots, \quad g_s = -\frac{3}{2\pi} b^{-2} \to 0, \quad (3.143)$$

justifying the prediction (3.121) of the TS/ST correspondence. Using the technique explained in Appendix B, we are able to compute 700 terms of this series, which allows us to perform in detail the resurgence analysis.

The integral form (3.141) of the normalised second trace can be evaluated explicitly by closing the contour of integral from above and summing up residues of poles. Following the exercise in Section 3.2.2, it is helpful to introduce artificially a "mass" deformation of the integral so that the integrand only has simple poles, and then turn off the mass deformation.

We consider the following integral

$$I(\xi, b) = \int_{\mathbb{R}} \frac{\sinh(2\pi h x)}{\sinh(\pi b x)} \frac{\Phi_b(x + \frac{\xi b}{2\pi} - 2ia + c_b) \Phi_b(x - \frac{\xi b}{2\pi} - 2ia + c_b)}{\Phi_b(x + \frac{\xi b}{2\pi} + 2ia - c_b) \Phi_b(x - \frac{\xi b}{2\pi} + 2ia - c_b)} e^{2\pi x(\frac{1}{3} b + b^{-1})} \mathrm{d}x. \quad (3.144)$$

We only consider the case $\xi$ is small and $\mathrm{Im}\, b^2 > 0$. Since $\mathrm{Re}\, b > 0$, the poles in the upper half plane are

$$ib^{-1} k, \quad \pm\frac{\xi b}{2\pi} + 2ia + irb + isb^{-1}, \quad k = 1, 2, 3, \dots, r, s = 0, 1, 2, 3, \dots \quad (3.145)$$

After summing up the residues at these poles, by using (A.6), we arrive at

$$\begin{aligned} I(\xi, b) =\ & ib^{-1} h_0(x, q) \tilde{h}_0(\tilde{x}, \tilde{q}) \\ & + ib\, w_a^{-1} \Big( h_1^+(x, q) \tilde{h}_1^+(\tilde{x}, \tilde{q}) - h_2^+(x, q) \tilde{h}_2^+(\tilde{x}, \tilde{q}) \\ & \qquad\qquad + h_1^-(x, q) \tilde{h}_1^-(\tilde{x}, \tilde{q}) - h_2^-(x, q) \tilde{h}_2^-(\tilde{x}, \tilde{q}) \Big), \end{aligned} \quad (3.146)$$

where we defined

$$x := e^{\xi b^2}, \quad \tilde{x} := e^{\xi}. \quad (3.147)$$

The holomorphic blocks are

$$h_0(x, q) = \frac{(q^{2/3} x; q)_\infty (q^{2/3} x^{-1}; q)_\infty}{(q^{1/3} x; q)_\infty (q^{1/3} x^{-1}; q)_\infty}, \quad (3.148a)$$

$$h_1^+(x, q) = q^{1/3} x \frac{(q; q)_\infty}{(q^{2/3}; q)_\infty} \frac{(q x^2; q)_\infty (q^{4/3} x; q)_\infty}{(q^{2/3} x^2; q)_\infty (q^{1/3} x; q)_\infty} {}_3\phi_2 \left( \begin{matrix} q^{2/3}, q^{2/3} x^2, q^{1/3} x \\ q x^2, q^{4/3} x \end{matrix}; q, q \right), \quad (3.148b)$$

$$h_2^+(x, q) = q^{2/9} x^{2/3} \frac{(q; q)_\infty}{(q^{2/3}; q)_\infty} \frac{(q x^2; q)_\infty (q^{4/3} x; q)_\infty}{(q^{2/3} x^2; q)_\infty (q^{1/3} x; q)_\infty} {}_3\phi_2 \left( \begin{matrix} q^{2/3}, q^{2/3} x^2, q^{1/3} x \\ q x^2, q^{4/3} x \end{matrix}; q, q^{2/3} \right), \quad (3.148c)$$

as well as

$$h_1^-(x, q) = h_1^+(x^{-1}, q), \qquad h_2^-(x, q) = h_2^+(x^{-1}, q). \quad (3.149)$$

The anti-holomorphic blocks are

$$
\tilde{h}_0(\tilde{x}, \tilde{q}) = \frac{(\tilde{q}w_a^{-1}\tilde{x}; \tilde{q})_\infty (\tilde{q}w_a^{-1}\tilde{x}^{-1}; \tilde{q})_\infty}{(w_a\tilde{x}; \tilde{q})_\infty (w_a\tilde{x}^{-1}; \tilde{q})_\infty}
$$
$$
\times \left( {}_3\phi_2\left( \begin{matrix} \tilde{q}, w_a\tilde{x}, w_a\tilde{x}^{-1} \\ \tilde{q}w_a^{-1}\tilde{x}, \tilde{q}w_a^{-1}\tilde{x}^{-1} \end{matrix}; \tilde{q}, \tilde{q}w_a^{-1} \right) - {}_3\phi_2\left( \begin{matrix} \tilde{q}, w_a\tilde{x}, w_a\tilde{x}^{-1} \\ \tilde{q}w_a^{-1}\tilde{x}, \tilde{q}w_a^{-1}\tilde{x}^{-1} \end{matrix}; \tilde{q}, \tilde{q} \right) \right), \quad (3.150a)
$$

$$
\tilde{h}_1^+(\tilde{x}, \tilde{q}) = \tilde{x}\frac{(\tilde{q}w_a; \tilde{q})_\infty}{(\tilde{q}; \tilde{q})_\infty} \frac{(\tilde{q}w_a\tilde{x}^2; \tilde{q})_\infty}{(\tilde{x}^2; \tilde{q})_\infty} {}_2\phi_1\left( \begin{matrix} w_a^{-1}, w_a^{-1}\tilde{x}^{-2} \\ \tilde{q}\tilde{x}^{-2} \end{matrix}; \tilde{q}, \tilde{q}w_a^{-1} \right), \quad (3.150b)
$$

$$
\tilde{h}_2^+(\tilde{x}, \tilde{q}) = \tilde{x}\frac{(\tilde{q}w_a; \tilde{q})_\infty}{(\tilde{q}; \tilde{q})_\infty} \frac{(\tilde{q}w_a\tilde{x}^2; \tilde{q})_\infty}{(\tilde{x}^2; \tilde{q})_\infty} {}_2\phi_1\left( \begin{matrix} w_a^{-1}, w_a^{-1}\tilde{x}^{-2} \\ \tilde{q}\tilde{x}^{-2} \end{matrix}; \tilde{q}, \tilde{q} \right), \quad (3.150c)
$$

as well as

$$
\tilde{h}_1^-(\tilde{x}, \tilde{q}) = \tilde{h}_1^+(\tilde{x}^{-1}, \tilde{q}), \qquad \tilde{h}_2^-(\tilde{x}, \tilde{q}) = \tilde{h}_2^+(\tilde{x}^{-1}, \tilde{q}). \quad (3.151)
$$

In the equations above,

$$
w_a = e^{-4\pi i a/b} = e^{-2\pi i/3}. \quad (3.152)
$$

The factorisation formula (3.146) can be further simplified. The holomorphic blocks $h_1^+(x, q)$ and $h_2^+(x, q)$ can also be expressed in terms of $q$-Appell functions, defined in (A.24),

$$
h_1^+(x, q) = q^{1/3}x\frac{(q^{5/3}; q)_\infty}{(q^{2/3}; q)_\infty}\Phi^{(1)}(q; q^{1/3}, q; q^{5/3}; q; q^{2/3}x^2, q^{1/3}x), \quad (3.153a)
$$

$$
h_2^+(x, q) = q^{2/9}x^{2/3}\frac{(q; q)_\infty(q^{4/3}; q)_\infty}{(q^{2/3}; q)_\infty^2}\Phi^{(1)}(q^{2/3}; q^{1/3}, q; q^{4/3}; q; q^{2/3}x^2, q^{1/3}x). \quad (3.153b)
$$

The summand of a $q$-Appell function $\Phi^{(1)}(a; b, b'; c; q; x, y)$ is $q$-holonomic. By using Takayama's algorithm [88, 89] of creative telescoping, implemented in the `HolonomicFunctions` package of Koutschan [90, 91], we find that $h_1^+(x, q)$, $h_2^+(x, q)$, together with the remaining three holomorphic blocks, satisfy a third order linear difference equation

$$
C_3(x; q)h(q^3x; q) + C_2(x; q)h(q^2x; q) + C_1(x; q)h(qx; q) + C_0(x; q)h(x; q) = 0, \quad (3.154)
$$

where

$$
\begin{aligned}
C_3(x; q) =& (1 - qx)(1 + qx)(1 - q^{5/3}x)(1 - q^{8/3}x)^2(1 + q^{8/3}x)(1 - qx^2)(1 - q^{13/3}x^2), \\
C_2(x; q) =& -q^{1/3}(1 - q^{5/3}x)(1 - q^2x)(1 + q^2x)(1 - qx^2) \\
& \left( 1 + q^{1/3} + q^{2/3} - q^{4/3}(1 - q^{1/3} + q^{2/3} + 2q)x \right. \\
& - (1 + q^{1/3})q^2(1 + q - q^{4/3} + 2q^2 - 3q^{7/3} + 2q^{8/3} + q^{10/3})x^2 \\
& + (1 + q^{1/3})q^4(1 + 2q^{2/3} - 3q + 2q^{4/3} - q^2 + q^{7/3} + q^{10/3})x^3 \\
& \left. + q^{22/3}(2 + q^{1/3} - q^{2/3} + q)x^4 - (1 + q^{1/3} + q^{2/3})q^9x^5 \right), \\
C_1(x; q) =& q(1 - qx)(1 + qx)(1 - q^{4/3}x)(1 - q^5x^2) \\
& \left( 1 + q^{1/3} + q^{2/3} - q^{4/3}(2 + q^{1/3} - q^{2/3} + q)x \right. \\
& - (1 + q^{1/3})q(1 + 2q^{2/3} - 3q + 2q^{4/3} - q^2 + q^{7/3} + q^{10/3})x^2 \\
& + (1 + q^{1/3})q^2(1 + q - q^{4/3} + 2q^2 - 3q^{7/3} + 2q^{8/3} + q^{10/3})x^3 \\
& \left. + q^{13/3}(1 - q^{1/3} + q^{2/3} + 2q)x^4 - (1 + q^{1/3} + q^{2/3})q^6x^5 \right), \\
C_0(x; q) =& -q^2(1 - q^{1/3}x)^2(1 + q^{1/3}x)(1 - q^{4/3}x)(1 - q^2x)(1 + q^2x)(1 - q^{5/3}x^2)(1 - q^5x^2).
\end{aligned}
$$
$$(3.155)$$

Therefore there are only three linearly independent holomorphic blocks. By making the ansatz that

$$h_{i_0}(x,q) = f_1(x,q)h_{i_1}(x,q) + f_2(x,q)h_{i_2}(x,q) + f_3(x,q)h_{i_3}(x,q), \tag{3.156}$$

where $h_{i_0}(x,q), h_{i_1}(x,q), h_{i_2}(x,q), h_{i_3}(x,q)$ are any four holomorphic blocks and the coefficients $f_j(x,q)$ are elliptic functions satsifying $f_j(qx,q) = f_j(x,q)$, we find

$$
\begin{aligned}
h_2^+(x,q) = & -\frac{q^{2/9}(q;q)_\infty^2\theta(-q^{-1/2}x;q)}{x^{1/3}\theta(-q^{-1/6};q)_\infty\theta(-q^{-1/6}x;q)}h_0(x,q) \\
& -\frac{x^{2/3}\theta(-q^{-1/6}x^2;q)}{q^{1/9}\theta(-q^{-1/2}x^2;q)}(h_1^+(x,q) - h_1^-(x,q)),
\end{aligned} \tag{3.157a}
$$

$$
\begin{aligned}
h_2^-(x,q) = & -\frac{q^{2/9}x^{1/3}(q;q)_\infty^2\theta(-q^{-1/2}/x;q)}{\theta(-q^{-1/6};q)\theta(-q^{-1/6}/x;q)}h_0(x,q) \\
& +\frac{\theta(-q^{-1/6}/x^2;q)}{q^{1/9}x^{2/3}\theta(-q^{-1/2}/x^2;q)}(h_1^+(x,q) - h_1^-(x,q)),
\end{aligned} \tag{3.157b}
$$

where we have used the notation

$$\theta(x;q) = (-q^{1/2}x;q)_\infty(-q^{1/2}/x;q)_\infty. \tag{3.158}$$

Notice that $h_1^+(x,q)$ and $h_1^-(x,q)$ always appear in these relations by the combination $h_1^+(x,q) - h_1^-(x,q)$, which is also true in the factorisation (3.146) once we take into account the fact that $\tilde{h}_1^-(\tilde{x},\tilde{q}) = -\tilde{h}_1^+(\tilde{x},\tilde{q})$. We can therefore write down a simplified factorisation of $I(\xi,\mathrm{b})$ by using only two reduced (anti-)holomorphic blocks

$$I(\xi,\mathrm{b}) = \mathrm{i}\mathrm{b}^{-1}h_0(x,q)\tilde{h}_0^{\mathrm{red}}(\tilde{x},\tilde{q}) + \mathrm{i}\mathrm{b}w_a^{-1}h_1^{\mathrm{red}}(x,q)\tilde{h}_1^{\mathrm{red}}(\tilde{x},\tilde{q}), \tag{3.159}$$

where

$$h_1^{\mathrm{red}}(x,q) = h_1^+(x,q) - h_1^-(x,q), \tag{3.160}$$

and

$$
\begin{aligned}
\tilde{h}_0^{\mathrm{red}}(\tilde{x},\tilde{q}) = & \tilde{h}_0(\tilde{x},\tilde{q}) + w_a^{-1}\frac{(\tilde{q};\tilde{q})_\infty^2\theta(-\tilde{q}^{1/2}\tilde{x};\tilde{q})}{\theta(-\tilde{q}^{1/2}w_a;\tilde{q})\theta(-\tilde{q}^{1/2}w_a^{-1}\tilde{x};\tilde{q})}\tilde{h}_2^+(\tilde{x},\tilde{q}) \\
& + w_a^{-1}\frac{(\tilde{q};\tilde{q})_\infty^2\theta(-\tilde{q}^{1/2}\tilde{x}^{-1};\tilde{q})}{\theta(-\tilde{q}^{1/2}w_a;\tilde{q})\theta(-\tilde{q}^{1/2}w_a^{-1}\tilde{x}^{-1};\tilde{q})}\tilde{h}_2^-(\tilde{x},\tilde{q}),
\end{aligned} \tag{3.161a}
$$

$$
\begin{aligned}
\tilde{h}_1^{\mathrm{red}}(\tilde{x},\tilde{q}) = & \tilde{h}_1^+(\tilde{x},\tilde{q}) - w_a\frac{\theta(-\tilde{q}^{1/2}w_a^{-1}\tilde{x}^2;\tilde{q})}{\theta(-\tilde{q}^{1/2}\tilde{x}^2;t q)}\tilde{h}_2^+(\tilde{x},\tilde{q}) \\
& + w_a\frac{\theta(-\tilde{q}^{1/2}w_a^{-1}\tilde{x}^{-2};\tilde{q})}{\theta(-\tilde{q}^{1/2}\tilde{x}^{-2};t q)}\tilde{h}_2^-(\tilde{x},\tilde{q}).
\end{aligned} \tag{3.161b}
$$

Finally, taking the massless limit $\xi \to 0$ and using the relation

$$\mathrm{Tr}'\rho_{\mathbb{P}^2}^2 = \sqrt{3}\mathrm{b}\frac{\Phi_\mathrm{b}(-\frac{\mathrm{i}\mathrm{b}}{6} - \frac{\mathrm{i}\mathrm{b}^{-1}}{2})^2}{\Phi_\mathrm{b}(\frac{\mathrm{i}\mathrm{b}}{6} + \frac{\mathrm{i}\mathrm{b}^{-1}}{2})^2}I(0,\mathrm{b}), \tag{3.162}$$

we find the following elegant factorisation formula for the normalised second trace

$$\mathrm{Tr}'\rho_{\mathbb{P}^2}^2 = \mathrm{i}\sqrt{3}\left(\tilde{H}_0(\tilde{q}) + 3\mathrm{b}^4H_1(q)\tilde{H}_1(\tilde{q})\right), \tag{3.163}$$

where the only non-trivial massless holomorphic block is

$$H_1(q) = q^{1/3} \frac{(q;q)_\infty^4 (q^{1/3};q)_\infty^2}{(q^{2/3};q)_\infty^5 (q^{4/3};q)_\infty} \, {}_3\phi_2 \begin{pmatrix} q^{1/3}, q^{2/3}, q^{2/3} \\ q, q^{4/3} \end{pmatrix}; q, q^{2/3} \end{pmatrix}, \tag{3.164}$$

and the massless anti-holomorphic blocks are

$$\widetilde{H}_0(\tilde{q}) = {}_3\phi_2 \begin{pmatrix} w_a, w_a, \tilde{q} \\ w_a^{-1}\tilde{q}, w_a^{-1}\tilde{q} \end{pmatrix}; \tilde{q}, w_a^{-1}\tilde{q} \end{pmatrix} - {}_3\phi_2 \begin{pmatrix} w_a, w_a, \tilde{q} \\ w_a^{-1}\tilde{q}, w_a^{-1}\tilde{q} \end{pmatrix}; \tilde{q}, \tilde{q} \end{pmatrix}$$
$$+ \frac{(\tilde{q};\tilde{q})_\infty^2 (\tilde{q}w_a;\tilde{q})_\infty^2}{(\tilde{q}w_a^{-1};\tilde{q})_\infty^4} \, {}_2\phi_1 \begin{pmatrix} w_a^{-1}, w_a^{-1} \\ \tilde{q} \end{pmatrix}; \tilde{q}, \tilde{q} \end{pmatrix}, \tag{3.165a}$$

$$\widetilde{H}_1(\tilde{q}) = \frac{2(\tilde{q}w_a;\tilde{q})^6}{(\tilde{q};\tilde{q})^3 (\tilde{q}w_a^{-1};\tilde{q})^3}. \tag{3.165b}$$

The simple factorisation formula (3.163) enables us to rapidly and accurately calculate the value of the normalised second trace for $g_s$ in the upper half plane. Through a high precision numerical calculation we find the following relationship between the normalised second trace and the Borel resummation of $\Phi_{0;2}(g_s), \Phi_{1;2}(g_s)$.

- For $g_s$ in sector $II$

$$\text{Tr}' \rho_{\mathbb{P}2}^2 = s_{II}(\Phi_{0;2})(g_s), \tag{3.166}$$

thus $\text{Tr}' \rho_{\mathbb{P}2}^2(b)$ furnishes a holomorphic lift of $s_{II}(\Phi_{0;2})(g_s)$.

- For $g_s$ in sector $I$

$$\text{Tr}' \rho_{\mathbb{P}2}^2 = s_I(\Phi_{0;2})(g_s) + s_I(\Phi_{1;2})(g_s) K_2(q), \tag{3.167}$$

where

$$K_2(q) = q^{1/6}(1 - 3q^{1/3} + 10q^{2/3} + \ldots). \tag{3.168}$$

By comparing these two identities we conclude immediately

$$\mathsf{S}_{01;2}^+(q) = K_2(q). \tag{3.169}$$

We comment that the two relations (3.166), (3.167) also justify the claim (3.143). When $g_s$ is in the second quadrant, (3.166) clearly implies (3.143). When $g_s$ is in the first quadrant, (3.167) indicates that in the leading order

$$\text{Tr}' \rho_{\mathbb{P}2}^2(b) \sim \Phi_{0;2}(g_s) + e^{-(3\mathcal{V}_{\mathbb{P}2} + \frac{i}{2})/g_s}(\ldots) \tag{3.170}$$

and the series $\Phi_{0;2}(g_s)$ clearly dominates.

In the next subsection we consider the radial asymptotic analysis of the anti-holomorphic blocks, which is a more efficient way to compute the $q$-series $\mathsf{S}_{01;2}^+$.

### 3.3.4 Radial asymptotic analysis for the second trace

We study here the representation of the anti-holomorphic blocks in terms of Borel resummation of asymptotic series. In general, we find

$$i\sqrt{3} \begin{pmatrix} \widetilde{H}_0(\tilde{q}) \\ -\frac{27}{4\pi^2 g_s^2} q^{1/3} \widetilde{H}_1(\tilde{q}) \end{pmatrix} = M_R(q) s_R \begin{pmatrix} \Phi_{0;2} \\ -\frac{1}{3} q^{1/6} \Phi_{1;2} \end{pmatrix}(g_s) \tag{3.171}$$

with

$$M_R(q) = \begin{pmatrix} 1 & M_{01}^R(q) \\ 0 & M_{11}^R(q) \end{pmatrix} \tag{3.172}$$

where $M_{01}^R(q), M_{11}^R(q)$ are $q$-series that parametrise non-perturbative corrections, and their expressions depend on the sector $R$ the coupling $g_s$ is in. The Borel resummation of $\Phi_{1;2}(g_s)$ is known exactly, and since we have 700 terms of $\Phi_{0;2}(g_s)$, we can compute the Borel resummation with very high numerical precision[11], and therefore we are able to compute the non-perturbative corrections to very high orders:

$$
\begin{aligned}
M_{01}^I(q) =& -2 + 3q^{1/3} - 6q^{2/3} + 11q - 18q^{4/3} + 27q^{5/3} - 37q^2 + 45q^{7/3} - 48q^{8/3} \\
& + 45q^3 - 36q^{10/3} + 24q^{11/3} - 18q^4 + \ldots, \tag{3.173a}
\end{aligned}
$$

$$
\begin{aligned}
M_{01}^{II}(q) =& 1 + 3q^{1/3} + 6q^{2/3} + 8q + 9q^{4/3} + 12q^{5/3} + 17q^2 + 18q^{7/3} + 15q^{8/3} + 19q^3 \\
& + 30q^{10/3} + 30q^{11/3} + 20q^4 + 27q^{13/3} + \ldots, \tag{3.173b}
\end{aligned}
$$

$$
\begin{aligned}
M_{11}^I(q) =& 1 - 6q^{1/3} + 18q^{2/3} - 35q + 42q^{4/3} - 9q^{5/3} - 85q^2 + 204q^{7/3} - 225q^{8/3} \\
& - 10q^3 + 522q^{10/3} - 990q^{11/3} + 775q^4 + \ldots, \tag{3.173c}
\end{aligned}
$$

$$
\begin{aligned}
M_{11}^{II}(q) =& 1 + 3q^{1/3} - 5q + 6q^{4/3} + 9q^{5/3} - 25q^2 - 3q^{7/3} + 63q^{8/3} - 45q^3 - 96q^{10/3} \\
& + 180q^{11/3} + 26q^4 + \ldots \tag{3.173d}
\end{aligned}
$$

It is easy to identify that

$$
M_{11}^I(q) = \frac{(q^{1/3};q)_\infty^6}{(q;q)_\infty^3 (q^{2/3};q)_\infty^3}, \quad M_{11}^{II}(q) = \frac{(q^{2/3};q)_\infty^6}{(q;q)_\infty^3 (q^{1/3};q)_\infty^3}. \tag{3.174}
$$

By inverting the matrix $M_R(q)$ in (3.171) one finds the holomorphic lift of $s_R(\Phi_{0;2})(g_s)$

$$
s_R(\Phi_{0;2})(g_s) = i\sqrt{3}\left( \widetilde{H}_0(\tilde{q}) + 3\mathsf{b}^4 \frac{M_{01}^R}{M_{11}^R} q^{1/3} \widetilde{H}_1(\tilde{q}) \right). \tag{3.175}
$$

Comparison with (3.166) together with (3.163) leads to

$$
M_{01}^{II}(q) = \frac{(q^{2/3};q)_\infty (q;q)_\infty}{(q^{1/3};q)_\infty (q^{4/3};q)_\infty} \, {}_3\phi_2\left( \begin{matrix} q^{1/3}, q^{2/3}, q^{2/3} \\ q, q^{4/3} \end{matrix}; q, q^{2/3} \right), \tag{3.176}
$$

which agrees with the numerical result (3.173b). Finally by comparing (3.171) applied in both sectors $I$ and $II$ we find

$$
\mathsf{S}_{01;2}^+(q) = K_2(q) = -\frac{1}{3}q^{1/6}\left( M_{01}^I(q) - \frac{M_{01}^{II}(q)}{M_{11}^{II}(q)} M_{11}^I(q) \right), \tag{3.177}
$$

which yields (3.139).

With the above ingredients, it should be possible to obtain appropriate descendants of the holomorphic blocks, as in [11,12] and section 3.2.2. This would make it possible to conjecture exact expressions for the $q$-series in (3.173), and therefore for the Stokes constants.

# 4 Conclusions

In this paper we have identified an infinite family of asymptotic power series arising naturally in topological string theory, by considering the conifold free energies and "quantizing" the flat coordinates. We have argued that these series are the analogues of conventional perturbative expansions in the coupling constant in finite rank gauge theories. We have conjectured that the

---

[11]We further improved the precision of Borel resummation with the help of conformal maps [92,93].

resurgent structure of these series are encoded in "peacock patterns", namely, infinite towers of singularities in the Borel plane with *integer* Stokes constants. The latter can then be regarded as new integer invariants of Calabi–Yau threefolds. We have verified our conjecture in various non-trivial toric examples by using the TS/ST correspondence. Thanks to this correspondence, the asymptotic series are promoted to spectral traces, which turn out to satisfy a factorization property into (anti)holomorphic blocks. This makes it possible to calculate the Stokes data in a very efficient way, and in some cases one obtains closed expressions for the Stokes constant as coefficients of explicit $q$-series. The resulting mathematical structure is very similar to the one found in complex CS theory.

Our results raise many questions, both technical and conceptual. Let us list some open problems and further directions to explore.

In our construction, it is crucial to enlarge the original perturbative series to a more general set of trans-series, in what we have called a minimal resurgent structure. It would be important to have a better understanding of the trans-series obtained in this way. In particular, it would be interesting to clarify whether they correspond to the conifold expansion of the general trans-series constructed in [29, 30]. More generally, we would like to have a more physical understanding of these trans-series, either in geometric terms (e.g. as contributions of D-brane sectors in the CY threefold) or in terms of a dual, quantum-mechanical description.

In this paper we have explored the conventional topological string theory side of the TS/ST correspondence, but one could also consider the semi-classical limit of the spectral traces $\hbar \to 0$. This is the dual story to what we analyzed here, since it involves an $S$-duality transformation of the string coupling constant, and it should make contact with the NS limit of the topological string.

We have also focused on the toric case for computational reasons, but as emphasized in section 2, our framework is very general and in particular it applies to compact CY examples, like e.g. the quintic CY. One can in principle use the genus expansion of topological string theory (as obtained in [94]) to extract perturbative series around the conifold point, and then study their Borel structure and Stokes constants. Although the existing data are probably not enough to perform a precise resurgent analysis, the point of view advocated in this paper might unveil new integrality structures in the compact case and shed light on the all-genus structure of topological string theory.

In our study, we have focused on the "numerical" series appearing after setting $\lambda_i = -N_i g_s$. As we explained, this leads to a simpler problem in the theory of resurgence, involving Gevrey-1 series with no parametric dependence (besides the "easy" one due to mass parameters). It is however natural to include the full dependence on the CY moduli. To do this, one could take as a starting point the full perturbative series of conifold free energies, $\mathcal{F}_g(\lambda)$, $g = 0, 1, \cdots$, and study the minimal resurgent structure associated to it, together with its Stokes constants. Some preliminary steps in that direction were taken in [60], where additional trans-series were obtained with the techniques of [29, 30], and used to reconstruct the fermionic spectral trace $Z_N(\hbar)$ for arbitrary $N$. There are many indications that, for each value of $\lambda$, the following formal series in $g_s$,

$$Z(\lambda; g_s) = \exp\left( \sum_{g \geq 0} \mathcal{F}_g(\lambda) g_s^{2g-2} \right), \tag{4.1}$$

will lead to peacock patterns, involving infinite towers of singularities in the Borel plane of $g_s$ (these towers are visible in the figures of [60], and in the simpler case of the resolved conifold they have been studied in [95]). It would be very interesting to understand these patterns and the resulting Stokes constants. Many questions arise naturally: Do the Stokes data depend on $\lambda$? Do they satisfy integrality properties? It is also likely that the Stokes constants obtained in this way (or by considering the similar problem for the NS free energies) are related to the

counting of BPS invariants in the CY threefold. This more general framework might lead to a better understanding of the enumerative meaning of the Stokes constants calculated in this work.

One could also consider, instead of the conifold free energies, the more conventional large radius free energies, and to study the Stokes constants for this different perturbative series. In the toric case, the TS/ST correspondence asserts that this series is the asymptotic expansion of the spectral determinant (3.1) in an appropriate scaling limit. It is also an open problem to "decode" this spectral determinant in terms of the Borel resummation of an appropriate set of trans-series at large radius.

The similarities with complex CS theory underlined in this work also suggest many questions: what is the $q$-difference equation satisfied by the spectral determinant $\Xi_X$? Is it possible to write $\Xi_X$ in terms of moduli-dependent, holomorphic blocks, similar to what was done in [12, 73]? The answers to these questions would shed new light on the structures appearing in the TS/ST correspondence, which are not yet fully understood.

# Acknowledgements

We would like to thank Jorgen Andersen, Tom Bridgeland, Bertrand Eynard, Rinat Kashaev, Maxim Kontsevich and Greg Moore for useful comments. We are specially grateful to Stavros Garoufalidis for many discussions on these topics and for collaboration in a related project. We also thank Claudia Rella for a detailed reading of the manuscript. This work has been supported in part by the Fonds National Suisse, subsidy 200020-175539, by the NCCR 51NF40-182902 "The Mathematics of Physics" (SwissMAP), and by the ERC-SyG project "Recursive and Exact New Quantum Theory" (ReNewQuantum), which received funding from the European Research Council (ERC) under the European Union's Horizon 2020 research and innovation program, grant agreement No. 810573.

# A  Quantum dilogarithm and $q$-series

## A.1  Faddeev's non-compact quantum dilogarithm

The quantum dilogarithm $\Phi_b(x)$ is defined in the strip $|\mathrm{Im}\,z| < |\mathrm{Im}\,c_b|$ as [96, 97]

$$\Phi_b(x) = \exp\left(\int_{\mathbb{R}+i\epsilon} \frac{e^{-2ixz}}{4\sinh(zb)\sinh(zb^{-1})}\frac{dz}{z}\right),\qquad (A.1)$$

which can be analytically continued to all values of b with $b^2 \notin \mathbb{R}_{\leq 0}$. When $\mathrm{Im}\,b^2 > 0$, the integral can be evaluated explicitly and one finds

$$\Phi_b(x) = \frac{(e^{2\pi b(x+c_b)};q)_\infty}{(e^{2\pi b^{-1}(x-c_b)};\tilde{q})_\infty},\qquad (A.2)$$

where

$$q = e^{2\pi i b^2},\qquad \tilde{q} = e^{-2\pi i b^{-2}},\qquad \mathrm{Im}(b^2) > 0\qquad (A.3)$$

and

$$c_b = \frac{i}{2}\left(b + b^{-1}\right).\qquad (A.4)$$

From this infinite product representation, one finds that $\Phi_b(x)$ is a meromorphic function of $x$ with

$$\text{poles: } x_{m,n} = c_b + imb + inb^{-1},\qquad \text{zeros: } -c_b - imb - inb^{-1},\quad m,n \in \mathbb{N}.\qquad (A.5)$$

The residues at poles can be found from the identity [75]

$$\Phi_b(x + x_{m,n}) = \frac{(q;q)_\infty}{(\tilde{q};\tilde{q})_\infty} \frac{1}{(q;q)_m} \frac{1}{(\tilde{q}^{-1};\tilde{q}^{-1})_n} \frac{\phi_m(2\pi b x)\widetilde{\phi}_n(2\pi b^{-1})}{1 - e^{2\pi b^{-1}x}} \tag{A.6}$$

where

$$\begin{aligned}
\phi_m(x) &= \frac{(q^{m+1}e^x;q)_\infty}{(q^{m+1};q)_\infty}, \\
\widetilde{\phi}_n(x) &= \frac{(\tilde{q};\tilde{q})_\infty}{(\tilde{q}e^x;\tilde{q})_\infty} \frac{(\tilde{q}^{-1};\tilde{q}^{-1})_n}{(\tilde{q}^{-1}e^x;\tilde{q}^{-1})_n}.
\end{aligned} \tag{A.7}$$

Furthermore, we have the expansion properties of $\phi_m(x), \widetilde{\phi}_n(x)$

$$\begin{aligned}
\phi_m(x) &= \exp\left(-\sum_{\ell=1}^{\infty} \frac{1}{\ell!} E_\ell^{(m)}(q) x^\ell\right), \\
\widetilde{\phi}_n(x) &= \exp\left(+\sum_{\ell=1}^{\infty} \frac{1}{\ell!} \widetilde{E}_\ell^{(n)}(\tilde{q}) x^\ell\right),
\end{aligned} \tag{A.8}$$

where

$$\begin{aligned}
E_k^{(m)}(q) &= \sum_{s=1}^{\infty} \frac{s^{k-1} q^{s(m+1)}}{1-q^s}, \\
\widetilde{E}_k^{(n)}(\tilde{q}) &= \begin{cases}
-n + E_1^{(n)}(\tilde{q}) & \text{if } k = 1 \\
E_k^{(n)}(\tilde{q}) & \text{if } k > 1 \text{ is odd} \\
2E_k^{(0)}(\tilde{q}) - E_k^{(n)}(\tilde{q}) & \text{if } k > 1 \text{ is even}
\end{cases}.
\end{aligned} \tag{A.9}$$

The expansion of $\Phi_b(x)$ near its zeros can be found using the inversion formula (A.10).

Some additional useful properties of quantum dilogarithms include

- Inversion formula

$$\Phi_b(x)\Phi_b(-x) = e^{\pi i x^2}\Phi_b(0)^2, \qquad \Phi_b(0) = \left(\frac{q}{\tilde{q}}\right)^{\frac{1}{48}} = e^{\pi i (b^2 + b^{-2})/24}. \tag{A.10}$$

- Complex conjugation

$$\Phi_b(x)^* = \frac{1}{\Phi_{b^*}(x^*)}. \tag{A.11}$$

- Special value

$$\Phi_b\left(\frac{i}{2}(b - b^{-1})\right) = \frac{(q;q)_\infty}{(\tilde{q};\tilde{q})_\infty} = b^{-1} e^{\frac{\pi i}{4} - \frac{\pi i}{12}(b^2 + b^{-2})}. \tag{A.12}$$

- Asymptotic behavior

$$\Phi_b(x) \sim \begin{cases}
\Phi_b(0)^2 e^{\pi i x^2} & \text{when } \operatorname{Re}(x) \gg 0, \\
1 & \text{when } \operatorname{Re}(x) \ll 0.
\end{cases} \tag{A.13}$$

- Quasi-periodicity

$$\frac{\Phi_b(x + c_b + ib)}{\Phi_b(x + c_b)} = \frac{1}{1 - qe^{2\pi bx}} \tag{A.14a}$$

$$\frac{\Phi_b(x + c_b + ib^{-1})}{\Phi_b(x + c_b)} = \frac{1}{1 - \tilde{q}^{-1}e^{2\pi b^{-1}x}}. \tag{A.14b}$$

- When b is small, we have the asymptotic expansion

$$\log \Phi_b\left(\frac{x}{2\pi b}\right) \sim \sum_{k=0}^{\infty} \left(2\pi i b^2\right)^{2k-1} \frac{B_{2k}(1/2)}{(2k)!} \mathrm{Li}_{2-2k}(-e^x), \tag{A.15}$$

where $B_{2k}(z)$ are Bernoulli polynomials. Similarly when b is large, we have

$$\log \Phi_b\left(\frac{x}{2\pi b^{-1}}\right) \sim -\sum_{k=0}^{\infty} \left(-2\pi i b^{-2}\right)^{2k-1} \frac{B_{2k}(1/2)}{(2k)!} \mathrm{Li}_{2-2k}(-e^x). \tag{A.16}$$

### A.2 Compact quantum dilogarithm

The compact quantum dilogarithm is defined by [97]

$$(qx; q)_\infty = \prod_{n \geq 1} (1 - xq^n), \qquad |q| < 1, \tag{A.17}$$

and is an entire function of $x$. It has an asymptotic expansion around $q$ a root of unity. The expansion around $q = 1$ is given by

$$\log(qx; q)_\infty = \sum_{n \geq 0} \frac{B_n(1)\hbar^{n-1}}{n!} \mathrm{Li}_{2-n}(x), \tag{A.18}$$

where

$$q = e^\hbar, \tag{A.19}$$

and $B_n(z)$ is the Bernoulli polynomial, defined by the generating function

$$\frac{e^{zt}}{e^t - 1} = \sum_{n \geq 0} B_n(z) \frac{t^{n-1}}{n!}. \tag{A.20}$$

In general, for arbitrary $s \in \mathbb{Z}$ and $x \neq 1$, one has

$$\log(q^s x; q)_\infty = \sum_{n=0}^{\infty} \hbar^{n-1} \frac{B_n(s)}{n!} \mathrm{Li}_{2-n}(x). \tag{A.21}$$

### A.3 $q$-series

The $q$-hypergeometric function is defined by

$$_{r+1}\phi_s\begin{pmatrix} a_0, & a_1, & \cdots, & a_r \\ & b_1, & b_2, & \cdots, & b_s \end{pmatrix}; q, z = \sum_{n=0}^{\infty} \frac{(a_0; q)_n (a_1; q)_n \cdots (a_r; q)_n}{(q; q)_n (b_1; q)_n \cdots (b_s; q)_n} \left((-1)^n q^{\binom{n}{2}}\right)^{s-r} z^n. \tag{A.22}$$

Heine's first transformation for the $(2, 1)$ $q$-hypergeometric series reads

$$_2\phi_1\begin{pmatrix} a, & b \\ & c \end{pmatrix}; q, z = \frac{(b; q)_\infty (az; q)_\infty}{(c; q)_\infty (z; q)_\infty} {_2\phi_1}\begin{pmatrix} c/b, & z \\ & az \end{pmatrix}; q, b. \tag{A.23}$$

The $q$-Appell function $\Phi^{(1)}(a; b, b'; c; q; x, y)$ is defined by

$$\Phi^{(1)}(a; b, b'; c; q; x, y) = \sum_{m,n \geq 0} \frac{(a; q)_{m+n}(b; q)_m (b'; q)_n}{(q; q)_m (q; q)_n (c; q)_{m+n}} x^m y^n. \tag{A.24}$$

# B  Fast algorithm for the second resurgent structure of local $\mathbb{P}^2$

In this section we present a fast algorithm to compute the asymptotic series $\Phi_{0;2}(g_s)$ for the geometry of local $\mathbb{P}^2$.

We find from (3.171) that the series $\Phi_{0;2}(g_s)$ appears in the asymptotic analysis of the anti-holomorphic block $\widetilde{H}_0(\tilde{q})$ given by (3.165a). This is actually also true for the partial anti-holomorphic block

$$\widetilde{H}_0^{\text{part}}(\tilde{q}) = {}_3\phi_2\left(\begin{matrix} w_a, w_a, \tilde{q} \\ w_a^{-1}\tilde{q}, w_a^{-1}\tilde{q} \end{matrix}; \tilde{q}, w_a^{-1}\tilde{q}\right) - {}_3\phi_2\left(\begin{matrix} w_a, w_a, \tilde{q} \\ w_a^{-1}\tilde{q}, w_a^{-1}\tilde{q} \end{matrix}; \tilde{q}, \tilde{q}\right). \tag{B.1}$$

We show that it is possible to extract the series $\Phi_{0;2}(g_s)$ from this partial anti-holomorphic block in the small $g_s$ limit in a very efficient way.

We first spell out the summation in the definition of the $q$-hypergeometric functions,

$$\widetilde{H}_0^{\text{part}}(\tilde{q}) = \sum_{k=0}^{\infty} \frac{(w_a; \tilde{q})_k^2}{(w_a^{-1}\tilde{q}; \tilde{q})_k^2} w_a^{-2k}\tilde{q}^k\left((-w_a^{-1/2})^k - (w_a^{1/2})^k\right), \tag{B.2}$$

where we used the property that $w_a^3 = 1, w_a^{3/2} = -1$. We denote the $\tilde{q}$-dependent component in the summand as

$$R(k; w_a) := \frac{(w_a; \tilde{q})_k^2}{(w_a^{-1}\tilde{q}; \tilde{q})_k^2} w_a^{-2k}\tilde{q}^k. \tag{B.3}$$

Let us introduce the notation

$$\tilde{q} = e^{\eta}, \quad \eta = \frac{4\pi^2 i}{3} g_s. \tag{B.4}$$

In the limit

$$g_s \to 0, \quad \eta \to 0 \tag{B.5}$$

we find with the help of (A.21) that

$$R(k; w_a) = \exp\left\{k(\eta - 2\log(w_a)) + 2\sum_{\ell=0}^{\infty} \eta^{\ell-1}\left(\frac{B_\ell(0) - B_\ell(k)}{\ell!}\text{Li}_{2-\ell}(w_a)\right.\right.$$
$$\left.\left. -\frac{B_\ell(1) - B_\ell(1+k)}{\ell!}\text{Li}_{2-\ell}(w_a^{-1})\right)\right\}. \tag{B.6}$$

By using the polylogarithm identitities

$$\begin{aligned} \text{Li}_{-n}(z) + (-1)^n\text{Li}_{-n}(1/z) &= 0, \quad n = 1, 2, \dots \\ \text{Li}_0(z) + \text{Li}_0(1/z) &= -1 \\ \text{Li}_1(z) - \text{Li}_1(1/z) &= -\log(-z), \quad z \notin (0, 1] \end{aligned} \tag{B.7}$$

and the Bernoulli polynomial identity

$$(-1)^\ell B_\ell(x) = B_\ell(1-x), \quad \ell \geq 0, \tag{B.8}$$

the expression of $R(k; w_a)$ can be simplified to

$$R(k; w_a) = \exp\left\{-k^2\eta + \sum_{\ell=2}^{\infty}\eta^{\ell-1}\frac{2B_\ell(0) - B_\ell(k) - B_\ell(-k)}{\ell!}(\text{Li}_{2-\ell}(w_a) + \text{Li}_{2-\ell}(w_a))\right\}. \tag{B.9}$$

If we expand this expression in terms of $\eta$, we find that the coefficients are polynomials of $k$; more precisely,

$$R(k; w_a) = 1 + \sum_{n=1}^{\infty}\eta^n\sum_{m=2}^{2n}R_{n;m}(w_a)k^m, \tag{B.10}$$

where we use $R_{n;m}(w_a)$ to denote the $w_a$-dependent coefficients in the $k$-polynomial. Finally by summing up the index $k$ and using the definition of the polylogarithm

$$\text{Li}_{-m}(x) = \sum_{k=0}^{\infty} k^m x^k, \tag{B.11}$$

we arrive at the following formula

$$\widetilde{H}_0^{\text{part}}(\tilde{q}) \sim -\frac{i}{\sqrt{3}} + \sum_{n=1}^{\infty} \eta^n \sum_{m=2}^{2n} R_{n;m}(w_a) \left( \text{Li}_{-m}(-w_a^{-1/2}) - \text{Li}_{-m}(-w_a^{1/2}) \right)$$

$$= -\frac{i}{\sqrt{3}} \left( 1 + \frac{8\pi^2}{9\sqrt{3}} g_s + \frac{16\pi^4}{81} g_s^2 + \cdots \right). \tag{B.12}$$

The first line here is strictly speaking not an identity because (B.11) is only valid for $|x| < 1$, while $|-w_a^{-1/2}| = |-w_a^{1/2}| = 1$. We also know from the radial asymptotic formula (3.171) that the first line of (B.12) cannot be possibly exact. The calculation from the first to the second line, however, is legitimate, and we identify the series to be $-i/\sqrt{3} \cdot \Phi_{0;2}(g_s)$. Therefore we cand use the first line of (B.12) to calculate $\Phi_{0;2}(g_s)$, which turns out to be very efficient, allowing us to obtain 700 terms of $\Phi_{0;2}(g_s)$.

## C Hunter-Guerrieri algorithm

The Hunter-Guerrieri algorithm [87] is concerned with the following problem. Let $p(\zeta)$ be a function analytic at the origin which has the following expansion

$$p(\zeta) = \sum_{n=0}^{\infty} p_n \zeta^n. \tag{C.1}$$

Let us suppose that the singularities of $p(\zeta)$ which are closest to the origin are located at $\zeta_\omega$ and $\zeta_\omega^*$, which are complex conjugate to each other. Suppose that, at these two singular points, $p(\zeta)$ has the following expansion

$$p(\zeta_\omega + \xi) = (-\xi/\zeta_\omega)^{-\nu} r(\xi) + \text{regular}, \tag{C.2a}$$

$$p(\zeta_\omega^* + \xi) = (-\xi/\zeta_\omega^*)^{-\nu} r^*(\xi) + \text{regular}, \quad \nu \notin \mathbb{Z}, \tag{C.2b}$$

where

$$r(\xi) = \sum_{k=0}^{\infty} b_k \xi^k. \tag{C.3}$$

Darboux' theorem [98, 99] implies that the large order behavior of $p_n$ is controlled by $b_k$ in the following manner

$$p_n \sim \sum_{k=0}^{\infty} \frac{2(-1)^k R^{k-n} \Gamma(n+\nu-k) B_k}{n! \Gamma(\nu-k)} \cos((n-k)\theta - \beta_k), \tag{C.4}$$

where we use the notation

$$\zeta_\omega = R e^{i\theta}, \quad b_k = B_k e^{i\beta_k}. \tag{C.5}$$

Imagine now that we have a long sequence of values of $p_n$. The problem is whether it is possible to extract $b_k$ from this sequence by using the formula (C.4).

The solution to this problem is of great interest for doing concrete calculations in the theory of resurgence introduced in Section 2.2. In that context, $p(\zeta)$ is the Borel transform $\widehat{\varphi}(\zeta)$ of certain Gevrey-1 series $\varphi(z)$, and $p_n$ are the coefficients $a_n/n!$. $r(\xi)$ is the Borel transform of the resurgent function $\varphi_\omega(z)$ at the singularity $\zeta_\omega$, and the coefficients $b_k$ are related to the $a_{n,\omega}$ appearing in the expansion (2.16) of $\varphi_\omega(z)$, by

$$b_k = \frac{\zeta_\omega^{-\nu}}{2i\sin\pi\nu} \frac{a_{k,\omega}}{\Gamma(k+1-\nu)}. \tag{C.6}$$

When two nearest singularities conjugate to each other are present in the Borel plane, the large order behavior of $a_n$ is given by (3.128) which is a consequence of (C.4). In this case, it is in general a difficult problem to extract information on $a_{k,\omega}$ from a sequence of $a_n$, as the usual method of Richardson extrapolation does not work.

Hunter and Guerrieri solved this problem beautifully in [87]. The first key step is to use the following composite and symmetric representation of $p(\zeta)$

$$p(\zeta) = \sum_{k=0}^{\infty} (c_k + d_k\zeta) \left[ (1-\zeta/\zeta_\omega)(1-\zeta/\zeta_\omega^*) \right]^{-\nu+k} + \text{regular}, \tag{C.7}$$

where the coefficients $c_k, d_k$ are real. We can relate the real coefficients $c_k, d_k$ to the complex coefficients $b_k$ by expanding (C.7) near $\zeta = \zeta_\omega$

$$p(\zeta) \sim \frac{e^{i\nu(\pi/2-\theta)}}{(2\sin\theta)^\nu} \left(1 - \frac{\zeta}{\zeta_\omega}\right)^{-\nu} \sum_{k=0}^{\infty} ((c_k + d_k\zeta_\omega) + d_k(\zeta-\zeta_\omega))$$
$$\cdot \sum_{\ell=0}^{\infty} \frac{(2i\sin\theta)^{k-\ell}\Gamma(k-\nu+1)}{\ell!\,\Gamma(k-\nu-\ell+1)} \left(\frac{\zeta-\zeta_\omega}{R}\right)^{k+\ell}, \tag{C.8}$$

and comparing coefficients with (C.2a). For instance, at leading order we have

$$c_0 + d_0\zeta_\omega = (2\sin\theta)^\nu e^{i\nu(\theta-\pi/2)} b_0. \tag{C.9}$$

In general, we have the relation

$$b_n = \frac{e^{i\nu(\pi/2-\theta)}}{(2\sin\theta)^\nu} R^{-n} \left[ (c_n + d_n\zeta_\omega)(2i\sin\theta)^n \right.$$
$$\left. + \sum_{k=0}^{n-1} \left( (c_k + d_k\zeta_\omega) \frac{(2i\sin\theta)^{2k-n}\Gamma(k+1-\nu)}{(n-k)!\,\Gamma(2k+1-n-\nu)} + Rd_k \frac{(2i\sin\theta)^{2k+1-n}\Gamma(k+1-\nu)}{(n-k-1)!\,\Gamma(2k+2-n-\nu)} \right) \right], \tag{C.10}$$

which can be used to relate the complex coefficient $b_n$ and the real coefficients $c_n, d_n$, whenever $R, \theta, \nu$, as well as $b_k, c_k, d_k$ for $k = 0, 1, \ldots, n-1$, are already known.

Now the key step is to rewrite (C.7) using the Gegenbauer polynomials, which are defined by the generating function

$$(1 - 2zs + s^2)^{-\sigma} = \sum_{n=0}^{\infty} C_n^\sigma(z) s^n. \tag{C.11}$$

We find

$$p_n \sim R^{-n} \sum_{k=0}^{\infty} c_k C_n^{\nu-k}(\mu) + R^{1-n} \sum_{k=0}^{\infty} d_k C_{n-1}^{\nu-k}(\mu), \quad \mu = \cos\theta. \tag{C.12}$$

This can be regarded as a convenient rearrangement of the asymptotic behavior (C.4). The Gegenbauer polynomials have very nice recursion relations. By exploiting cleverly these recursion relations we can define recursively the following objects,

$$S_n^0(R, \mu, \nu) = p_n, \tag{C.13}$$

$$S_n^{m+1}(R, \mu, \nu) = R^2 S_n^m(R, \mu, \nu) - \frac{2(n + \nu - 2m - 1)R\mu}{n} S_{n-1}^m(R, \mu, \nu)$$
$$+ \frac{(n + 2\nu - 3m - 2)(n - m - 1)}{n(n-1)} S_{n-2}^m(R, \mu, \nu), \tag{C.14}$$

so that in the asymptotics of $S_n^m$ the coefficients $c_k, d_k$ for $k = 0, \ldots, m-1$ have been canceled, and $S_n^m$ is progressively smaller with increasing $m$ and large fixed $n$. In fact, one can show

$$S_n^m(R, \mu, \nu) \sim \mathcal{O}(n^{\nu - 2m - 1}). \tag{C.15}$$

Then we can solve for the three unknowns $R, \mu, \nu$ approximately by three equations

$$S_n^m(R, \mu, \nu) = S_{n-\ell}^m(R, \mu, \nu) = S_{n-2\ell}^m(R, \mu, \nu) = 0. \tag{C.16}$$

The step $\ell$ is usually taken to be 1. The approximation becomes increasingly more accurate with larger $m$ and $n$.

There are several comments. First of all, in practice the equations (C.16) are non-linear equations. It is more convenient to solve them iteratively by the Newton–Raphson method. The derivatives $\partial_R S_n^m$, $\partial_\mu S_n^m$, $\partial_\nu S_n^m$ can be constructed recursively alongside $S_n^m$ with (C.13), (C.14). We take the solutions to $S_n^1$ as initial estimates for the Newton-Raphson method. To quickly solve $S_n^1$, we take four equations

$$S_n^1 = S_{n-\ell}^1 = S_{n-2\ell}^1 = S_{n-3\ell}^1 = 0, \tag{C.17}$$

and solve them as linear equations for four independent unknowns $R^2, R\mu, R\mu\nu, \nu$. The estimates for $R, \mu$ are then obtained by $R = \sqrt{R^2}, \mu = R\mu/\sqrt{R^2}$. Note that since these are approximate solutions they usually do not satisfy the relation

$$R\mu\nu = R\mu \cdot \nu. \tag{C.18}$$

Second, the Hunter-Guerrieri algorithm can be slightly modified to calculate coefficients $b_k$ as well. Once $R, \mu, \nu$ are known, we move $c_0, d_0$ to the left hand side in (C.12)

$$p_n^{(1)} \sim R^{-n} \sum_{k=0}^{\infty} c_{k+1} C_n^{\nu - 1 - k}(\mu) + R^{1-n} \sum_{k=0}^{\infty} d_{k+1} C_{n-1}^{\nu - 1 - k}(\mu), \tag{C.19}$$

where

$$p_n^{(1)} := p_n - R^{-n} c_0 C_n^\nu(\mu) - R^{1-n} d_0 C_{n-1}^\nu(\mu). \tag{C.20}$$

This indicates that we can do the same construction as (C.13), (C.14) with $p_n$ replaced by $p_n^{(1)}$ and $\nu$ reduced by 1. The $S_n^m(c_0, d_0)$ constructed in this way are functions of $c_0, d_0$, and in their asymptotics the coefficients $c_{k+1}, d_{k+1}$ for $k = 0, \ldots, m-1$ have been canceled so that they are progressively smaller with increasing $m$ and large fixed $n$. We can similarly solve the two unknowns $c_0, d_0$ with two equations

$$S_n^m(c_0, d_0) = S_{n-\ell}^m(c_0, d_0) = 0. \tag{C.21}$$

Note that $S_n^m(c_0, d_0)$ are linear in $c_0, d_0$, so (C.21) can be solved directly without resort to the Newton–Raphson method. In general, if we know $R, \mu, \nu$ as well as $c_j, d_j$ for $j = 0, 1, \ldots, k-1$, the coefficients $c_k, d_k$ can be solved from

$$S_n^m(c_k, d_k) = S_{n-\ell}^m(c_k, d_k) = 0. \tag{C.22}$$

Here, $S_n^m(c_k, d_k)$ are constructed by using (C.13), (C.14), where $\widehat{a}_n$ are replaced by

$$p_n^{(k)} := p_n - R^{-n} \sum_{j=0}^{k} c_j C_n^{\nu-j}(\mu) - R^{1-n} \sum_{j=0}^{k} d_j C_{n-1}^{\nu-j}(\mu) \tag{C.23}$$

and $\nu$ shifted to $\nu - k - 1$. Once $c_k$, $d_k$ are known, $b_k$ can be obtained by (C.10).

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
