# Peer review of "Peacock patterns and new integer invariants in topological string theory"

_SciPost Physics, doi:SciPost Phys. 12, 058 (2022)_

## Round 3 · Referee Report · Anonymous (Referee 1) · 2022-1-11

Strengths

  1. This paper presents interesting novel results in an area of intense current research.
  2. The authors define a new family of intrinsically non-perturbative numerical invariants for CY3-folds, conjectured to be integers and verified in examples.
  3. The authors show in detail how to derive these invariants in examples using the TS/ST correspondence, discussing the resulting mathematical structure and relating this to a counterpart in complex Chern-Simons theory.
  4. The paper is well structured and presents results clearly, nicely discussing them in a broader context. It furthermore strikes a good balance when providing background information and references.

Report

This paper builds on results from a series of recent works by the authors and their collaborators. Although rather technical, it is very well structured and presents interesting novel results clearly.

The authors introduce a new family of numerical invariants for Calabi-Yau 3-folds X, which they define in Section 2 as Stokes constants relying only on the resurgence structure of certain associated formal power series. These series are the conifold free energies $\mathcal{F}_g(\boldsymbol{\lambda})$ which are determined by topological string theory on X. They enter the partition function $\Phi_{\mathbf{N}}(g_s)$, which is a formal power series in the string coupling constant $g_s$ labeled by a tuple of integers $\mathbf{N}$ when viewing $\boldsymbol{\lambda}$ as 't Hooft parameters. From $\Phi_{\mathbf{N}}(g_s)$, the authors define an associated family of formal power series called the "minimal resurgent structure" $\mathcal{B}_{\Phi_{\boldsymbol{N}}}$, whose Borel transforms in turn define the Stokes constants associated to singularities in the Borel plane. The theory of resurgence allows to determine these invariants, which the authors conjecture to be integers and naturally organised into q-series.

This conjecture is verified in examples in Section 3, which presents a detailed analysis of these invariants in cases where X is toric, describing an efficient computational method through the TS/ST correspondence. The authors provide in this section a short review of this correspondence and discuss interesting parallels to results obtained in complex Chern Simons theory. The paper concludes with a concise summary in Section 4, which also provides a list of open problems . The three appendices collect the technical background for relevant special functions and computational techniques, while the introduction provides an excellent overview which sets this work in the broader context and ties together the different sections.

Requested changes

None

---

## Round 3 · Referee Report · Anonymous (Referee 2) · 2022-1-16

Report

This paper studies resurgent structures of topological string theory.
The starting point is the partition function of closed topological strings on toric Calabi-Yau threefolds near conifold points. Adopting the gauge/string large N duality dictionary, the Calabi-Yau Kahler moduli are interpreted as 't Hooft parameters for a gauge theory $\lambda_i\sim N_i g_s$.
For each value of $N_i$, the paper considers the Borel summation of the partition function, and studies its Stokes phenomena. A general lesson that emerges, is that singularities in the Borel plane occur in towers resembling `peacock patterns', whose number depends on $N_i$.
The paper presents precise conjectures on the type of `minimal resurgent structures' appearing in this context, concerning the number and types of asymptotic series associated to each singularity, and the Stokes matrices.
A remarkable conjecture formulated in this paper states that Stokes coefficients are integer. This is verified with rather nontrivial computations for local surfaces $\mathbb{F_0}$ and $\mathbb{P}^2$, for different values of $N_i$.
The findings of this paper are genuinely new, and suggest the existence of extensive new structures governing nonperturbative sectors of topological strings. This picture is supported by solid evidence, at least for a few examples.
These results and conjectures raises several interesting questions for future studies, concerning the interpretation of these structures and their integrality properties.

---

## Editorial Decision

published